# There and back again, a journey of many pathways: conceptualising the marine organic carbon cycle

Maike Iris Esther Scheffold[1] and Inga Hense[1]

[1]Institute of Marine Ecosystem and Fishery Science, Center for Earth System Research and Sustainability, University of Hamburg, Hamburg, Germany

**Correspondence:** Maike Iris Esther Scheffold (maike.scheffold@uni-hamburg.de)

**Abstract.** Understanding and determining the pathways that organic carbon (OC) takes in the ocean is one of the pressing tasks of our time, as the fate of OC in the ocean is linked to the climate system and the functionality of marine ecosystems. The multitude and complexity of these pathways are typically investigated with sophisticated, mainly quantitative methods focusing on individual pathways to resolve their interactions and processes as realistically as possible. In addition to these approaches to understand and recreate complexity, there is a need to identify commonalities and differences between individual OC pathways and define their overarching structures. Such structures can provide a framework for the growing number of partly overlapping concepts, which conceptualise selected OC pathways, and promote more systematic comparisons and consistent communication, especially between different disciplines. In response, we propose a (visual) concept in which we define such higher-level 'structures' by comparing and condensing marine OC pathways based on their sequences of processes and the layers of the marine system in which they operate. The resulting structures comprise 'closed loops', three remineralisation and two recalcitrant dissolved organic carbon loops that close in marine systems, and 'open loops', condensing pathways leaving the marine system to the atmosphere or deeper sediment layers. In addition, we provide a synthesis of embedded processes, OC pools, and process-executing organisms (agents) embedded in these loops. By translating a definition of the biological carbon pump into our concept, we show how the application and discussion of our defined structures facilitate a consistent visualisation, a systematic comparison of differently resolved concepts and studies, and integration of these in the larger picture of the marine OC cycle. As a complement to quantitative studies and descriptions of individual pathways, our concept decomposed the complexity of OC pathways by defining new universal structures. These structures provide a skeleton that can be adapted to different systems and filled with life by the users.

## 1  Introduction

The pathways along which organic carbon (OC) moves through oceanic systems affect not only the climate system (Barange et al., 2017) and ecosystem functioning (Griffiths et al., 2017), but also human well-being and socio-ecological systems (Ullah et al., 2018). Therefore, understanding marine OC pathways and the current and future marine OC dynamics resulting from the multiplicity of these pathways is an essential and very productive focus of ocean research (Jiao et al., 2018). Comprehensive observations and sophisticated numerical models, e.g. by the Joint Global Ocean Flux Study (JGOFS) (Doney and Ducklow,

2006), improved carbon budgets (e.g. by Giering et al. (2014)) and quantitative estimates of the contribution of individual organisms (e.g. in Bianchi et al. (2021)), to name but a few, are continuously expanding our understanding of OC pathways and the marine OC cycle.

Complementing the often-quantitative results, these studies sometimes provide (visual) concepts that abstractly describe and generalise OC pathways as a sequence of processes or a core mechanism. Due to the multitude of disciplines involved, the heterogeneity of ocean systems and the complexity of the marine OC cycle, these concepts often only consider a selection of pathways related to the respective research focus. For example, some studies conceptualise and generalise pathways for specific carbon pools e.g. dissolved OC in the microbial pump (Jiao et al., 2010; Jiao and Zheng, 2011), for a selection of species such as bacteria in the microbial loop (Azam et al., 1994) or for physical processes of different scales e.g. large-scale or eddy-subduction export (Levy et al., 2013; Omand et al., 2015).

The different foci and the limited spectrum of the pathways considered lead to concepts that complement each other (focusing on different processes or pools), but also promote partly overlapping sub-concepts. An example is the generalisation of pathways leading to the biota-induced vertical gradient of dissolved inorganic carbon in the oceans, described by the concept of the biological carbon pump (BCP). Several sub-concepts of the BCP have emerged, describing, among other things, the transport of carbon into and out of specific water layers, such as the mixed layer pump (Gardner et al., 1995), or carbon export by species-specific behaviour, such as the lipid pump (Jónasdóttir et al., 2015). Recent approaches to further generalise the pump concept by defining its main functions, e.g. particle injection by Boyd et al. (2019), show the need to define structural elements to make concept such as the BCP more comparable, comprehensive, systematic and adaptable.

It is plausible and useful that studies on individual OC pathways or systems produce specific and small-scale sub-concepts. However, in science, there is an additional need to identify commonalities and to find and define basic unifying structures (Scheiner and Willig, 2011). So far, no attempt has been made to summarise and generalise the OC pathways and conceptual ideas into an overarching general concept that represents structures of the marine OC cycle.

Existing concepts, especially those aiming at a more comprehensive representation of the marine OC cycle, are often not visually congruent within the respective graphics or compared to schemata in other publications. Processes and pathways are for instance not represented with the same level of detail. For example, Steinberg and Landry (2017), Cavan et al. (2019), Anderson and Ducklow (2001) and Boscolo-Galazzo et al. (2018) visually detach processes from their products, such as DIC, or do not mention some products in the figures at all. As the aim of such studies is not to create congruent conceptual representations of the marine OC cycle, their visualizations are still useful tools to highlight their research focus in an overarching picture. However, we would like to emphasise that graphics are a visualisation of the mapper's mental concepts. By deciding what to visualise and at what resolution, and by omitting information, parts of this mental concept are obscured, which can make it difficult to understand and use the concepts for studies other than the one for which it was created. Graphics are powerful tools for disseminating information, displaying concepts and promoting discussion (Margoluis et al., 2009). Non-congruent graphics do not exploit that full potential.

The lack of an overarching (and congruently visualised) concept of the marine OC cycle can reduce the transparency of the scientific process and make comparisons and discussions as well as the adaptation of concepts and ideas more difficult (Scheiner

and Willig, 2011). Different resolutions and definitions of pathways and overarching structures risk misunderstanding and mis-communication in education (Fortuin et al., 2011), among young but also more experienced researchers or in interdisciplinary communities (Heemskerk et al., 2003) and may foster a growing number of sub-concepts (Scheiner and Willig, 2011), some of which may overlap.

To reduce this risk, we propose to step back from quantitative, specific, and numerically advanced research and to summarise and generalise what is known about the marine OC cycle and pathways. The result of this step is a general concept that does not represent specific carbon processes or a single pathway but defines common structures of all pathways. We define these structures in linguistic and visual units by comparing and condensing similarities of possible OC pathways in the marine system. The result is the definition of several structures of 'closed' and 'open OC loops' that include all pathways that close within the marine system or leave the system into the deeper sediment or atmosphere.

The resulting concept facilitates 1) comparing models and concepts of different resolutions, 2) synthesizing concepts, definitions and scientific languages, 3) adding new scientific knowledge in a congruent and structured way, 4) identifying research gaps and inconsistencies, and 5) placing finite pathways into an overarching framework of the marine OC cycle. In this way, the concept can help researchers from different disciplines to facilitate research design, discuss individual concepts, and improve interdisciplinary communication, collaboration, and scientific education.

In the following, we describe how we developed our concept based on the questions 1) What are the different pathways for an OC compound in marine systems? 2) Which structures can be summarised? 3) Which processes, pools and agents are embedded in these structures? By answering the first two questions, we obtain a concept of universal structures of marine OC pathways. The last question allows us to identify the processes, pools and agents embedded in these structures, which allow defining smaller-scale structures that can be adapted to specific research questions and marine systems. In the discussion, we describe as an application example how a definition of BCP can be translated into our concept, and discuss the add-ons of this representation.

## 2 Concept specifications

Given that we conceptualise only the OC pathways (for a definition of relevant terms of the concept, see Table 1), we do not resolve carbonate and alkalinity interactions, and do not display marine carbonate systems within our concept.

In addition, we focus on OC that remains within the marine system, i.e. the water column plus upper sediment that still interacts with the water column. Therefore, we only consider pathways that start as OC within the surface waters, acknowledging that this initial position (Table 1) is an artificial construct since cycles do not start (or end) somewhere and marine carbon may originate from terrestrial run-off, atmospheric deposition, or photosynthesis. As soon as an OC pathway leaves the marine system, either into the atmosphere or into deeper sediment layers that do not interact with the water column, we not detailed describe them within this concept and assign them to 'open' loops. These loops close too, but outside our focal marine system.

It is irrelevant for our concept how much time an organic compound spends on the pathway. As such we are not interested in resolving the time scales of pathways and the accumulation of OC, standing stocks, in the system. Thus, it is the same pathway

when OC remains in the standing stock of a whale throughout its life and is respired at the surface right before its death and when OC is respired by a whale at the water surface immediately after being consumed. However, we do implicitly include time scales of pathways, since we consider different spatial scales closely connected to temporal scales (Dickey, 1990).

We provide a qualitative concept and are not interested in the amount of carbon that passes through the different pathways or the probabilities of OC to do so. We consider all pathways to being equally possible by assuming that each carbon compound finds the conditions for each pathway at the same time. For instance, the system provides suitable consumers that reduce sinking of material and at the same time a spatio-temporal mismatch with consumers that favours sinking.

For identifications of structures on a higher resolution (Sect. 3.2), we operationally subdivide OC into different pools, if the pathways involve OC of different size, volatility and lability. In such cases, we distinguish particulate organic carbon (POC), embedding living and non-living OC with sizes larger 0.2 $\mu$m (Kharbush et al., 2020), aggregates and marine snow; dissolved organic carbon (DOC), defined as non-living carbon smaller 0.2 $\mu$m (Kharbush et al., 2020); and volatile organic compounds (VOCs), such as dimethyl sulphate and $CH_4$. In addition, we separately consider recalcitrant (or refractory) DOC (rDOC), defined here as DOC that is remineralised on time scales between 1.5 and 40,000 years for semi-labile to ultra-refractory (Hansell, 2013), as opposed to 0.001 years for labile DOC (Hansell, 2013). We consider rDOC separately from DOC because rDOC is considered the only form of OC that accumulates in the water column in quantities relevant to the climate system (Jiao et al., 2010, 2011). We also include dissolved inorganic carbon (DIC) as an intermediate pool. Whilst this DIC pool consists of various inorganic (IC) molecules, we do not distinguish them within our concept.

## 3   A (visual) concept of the marine organic carbon cycle

### 3.1   Main structures of the marine organic carbon cycle

Our concept is based on the comparison and condensation of possible OC pathways using state-of-the-art knowledge. To this end, we generate a literature-based pathway concept (see Supplement A) by collecting and mapping the different pathways that an OC compound can "go" within the marine OC cycle based on a non-systematic literature review. The individual pathways in this concept are defined by *sequences of processes* (Table 1), such as sinking and remineralisation, and either return to the initial position in the surface waters or leave the marine system to the sediment or the atmosphere. We compare the OC pathways in the literature-based pathway concept and condense their similarities into generally applicable structures.

The structures, e.g. closed loops, are stripped of any processes, pools or involved agents (definitions see Table 1). We add this information in the next step (Sect. 3.2) allowing the definition of additional structures of higher resolution.

To explain how the pathways of the literature-based pathway concept can be compared and condensed to define structures of the marine OC cycle, we use as an analogy a town with a sandbank separated by a lagoon. The inhabitants of the town regularly visit the sandbank to spend their evenings at the beach. A route planner, comparable to our literature-based pathway concept, shows 100 individual pathways that end at the beach. These pathways are similar, but all differ in the overall sequence of streets and vehicles used.

**Table 1.** Definitions and examples of relevant terms based on three individual pathways.

| Example pathways in the literature-based pathway concept | **Pathway 1**: Phytoplankton DOC exudation → Bacterial remineralisation → DIC uptake by phytoplankton<br>**Pathway 2**: Zooplankton grazing on phytoplankton → Zooplankton respiration → DIC uptake by macrophytes<br>**Pathway 3**: Phytoplankton respiration → DIC outgassing |
|---|---|

| **Term** | **Definition** | **Examples based on pathways 1-3** |
|---|---|---|
| **Pool** | A reservoir of a certain substance, in this case organic carbon. Pools can be living and non-living. | Phytoplankton, DOC, Bacteria, DIC, Zooplankton, Macrophytes |
| **Agent** | An organism that initiates or executes a process. | Phytoplankton, Bacteria, Zooplankton, Macrophytes |
| **Space** | A spatially bounded volume with different environmental conditions. | Surface layer space (SLS), Atmosphere space (AS) |
| **Initial position** | Abstract start position of the OC pathways (OC in the SLS). | OC in the surface layer space (SLS) |
| **Process** | A self-contained change in the properties or position of carbon. A process is embedded in a functional segment. | Phytoplankton DOC exudation, Zooplankton grazing on phytoplankton, Bacterial remineralisation, Zooplankton respiration, Phytoplankton respiration, DIC uptake by phytoplankton, DIC uptake by macrophytes, DIC outgassing |
| **Functional segment** | The condensed function of processes that have the same general functionality. They are defined by the abstracted result of the processes, independent of species involved, etc. Functional segments comprise all globally applicable processes having the same general functionality. | OC size change, POC consumption, OC remineralisation, DIC uptake by primary producers, DIC exit |
| **Pathway** | An individual sequence of processes. The sequence can be translated to a sequence of functional segments. Each pathway is embedded in a structure. Although pathways can be described by sequences of functional segments, they always represent individual features and not condensed ones. | Pathway 1: OC size change → OC remineralisation → DIC uptake by primary producers<br>Pathway 2: POC consumption → OC remineralisation → DIC uptake by primary producers<br>Pathway 3: OC remineralisation → DIC exit |

125

| | | |
|---|---|---|
| **Structure** | A structure is a superordinate generalisation/ condensation of multiple pathways. A structure is defined by a unique combination of a sequence of functional segments and the involved spaces.<br><br>Syntax: Functional segment [Space]<br><br>Sequences of functional segments must be true for all pathways within the structure. Depending on the resolution, different structures can be defined. Structures can always be related to each other, e.g. the surface layer remineralisation loop is part of the superordinate structure closed loop (see below). | Structure 1: OC remineralisation [SLS] → DIC uptake by primary producers [SLS]<br><br>Structure 2: OC remineralisation [SLS] → DIC exit [AS] |
| **Closed loops** | A structure that comprises all pathways returning to the initial position is named closed loops. Closed loops are the most overarching structure in the marine OC cycle. | Surface layer remineralisation loop (SLRL): OC remineralisation [SLS] → DIC uptake by primary producers [SLS] |
| **Open loops** | A structure that comprises all pathways not returning to the initial position is named 'open' loops. | Atmospheric IC loop (AICL): OC remineralisation [SLS] → DIC exit [AS] |
| **Marine OC cycle** | The marine OC cycle consists of all closed loops. | |

There is, however, a common denominator for all pathways. To reach the beach, the lagoon must be crossed. This condition is independent of the way of crossing. People reach the sandbank in different ways, e.g. by public ferry or private boat. The result 'people reach the sandbank' and the general functionality 'crossing the lagoon' of these processes coincide. Therefore, we define 'crossing the lagoon' as a functional segment (*summarised function of the involved processes with the same general functionality*, table 1) common to all pathways to the beach. It should be noted that this does not mean that all pathways ONLY need this functional segment. The functional segment 'crossing the lagoon' is at least required to reach the beach and the bottleneck of ALL beach pathways.

At a higher resolution, which allows more complexity, differences of the beach pathways can be identified and grouped by defining the functional segments shared by these groups. For instance, people who do not live at the harbour front (functionality: 'living behind the harbour front') use one of three roads to reach the harbour (functionality: 'reaching the harbour'). These pathways share the sequence of the functional segments 'living behind the harbour front', 'reaching the harbour' and 'crossing the lagoon'. People living at the harbour front (functionality: 'living at the harbour front') only have to cross the lagoon and share the sequence of 'living at the harbour front' and 'crossing the lagoon'.

Based on these distinct sequences of functional segments, different structures can be defined. The most general and superordinate structure is the 'entire city-beach' structure defined by the functional segment 'crossing the lagoon', which is common

to all pathways. That sequence is the minimum sequence shared by all pathways and defines the highest-level structure. At the higher resolution, a distinction can be made between a 'harbour front-beach' structure (living at the harbour-front and crossing the lagoon) and a 'behind the harbour-beach' structure (living behind the harbour-front, reaching the harbour and crossing the lagoon). The sequences of functional segments minimally describe all pathways within these structures.

The resolution and thus the definition of structures is a matter of choice. One could for example also distinguish other structures based on the method of crossing the lagoon or find further differences and commonalities between the pathways in the rest of the city and define additional structures. However, assuming that the rest of the city has a very diverse and complicated road network, the 'harbour front-beach' and the 'behind the harbour-beach' structure may be sufficient to define, for instance, pressure points and bottlenecks when construction works block the three streets to the harbour.

Similarly to the description above, we define structures of the marine OC pathways based on the literature-based pathway concept (see Supplement B for a schematic of the methodological steps). One structure that immediately catches the eye are pathways that loop inside or outside the marine system. We define these structures as closed and 'open' loops. The closed loops are the highest structure in the marine OC cycle and the focus of this study. In the following, we define general structures hierarchically below the closed loops by comparing the pathways of these loops as described above. The structures we want to define should be as general as possible while still covering relevant differences.

We identify six functional segments that are necessary to describe the desired structures (Figure 1): *OC position change (A), Formation of rDOC (B), rDOC conversion to DOC (C), OC remineralisation (D), DIC upward position change (E) and DIC uptake by primary producers (F)*. We recognise that excluding the rDOC-related functional segments would further reduce the number of functional segments and structures. However, as described earlier, rDOC is relevant to the climate system and is related to very different phenomena and processes. So although it may not technically be the minimum solution, it is the minimum solution that still captures relevant differences.

The depths OC reaches on its pathways is another of these relevant difference that we want to resolve, as these depths affect the function of OC in the ecosystem (e.g. as a food source for benthic organisms), the environmental conditions it encounters (e.g. bioturbation) and the time it takes to return to the surface layer (e.g. years or decades). However, the functional segment OC position change (A) does not provide information on whether the position change ends in the water column or in the sediment.

Hence, to unambiguously define structures that account for the differences described above we need to add spatial information. To systematically add this information, we define five spaces, volumes with distinctly different environmental conditions and processes. After general considerations of the ocean layers, the *surface layer space (SLS)* encounters sufficient light to support photosynthesising organisms and primary production. Seasonal and continuous mixing counteract material loss and keep matter close to remineralisers. In the *water column space (WCS)* below the well-mixed layer, mixing occurs less frequently, more slowly or very infrequently, depending among other things on the water depth (DeVries et al., 2012). Matter takes more time to resurface and may escape remineralisers due to changing positions or its recalcitrant or degraded character (Baker et al., 2017). In the *upper sedimentation space (USS)*, remineralisers also remineralise highly degraded material as it remains in their vicinity longer than in the water column (Middelburg, 2019). The *lower sedimentary space (LSS)* is largely abiotic and

undisturbed and allows lithification processes. In addition, we define the *atmospheric space (AS)* above the marine system. The use and choice of spaces depend on the intended resolution of the structures. Users of the concept can change the spaces, e.g. by subdividing the water column space, resulting in a different number of closed loops, or omit the spatial extent completely if

they aim for an even more general description than ours. However, if the minimum number of closed loops is to be conceptually described at the same level of resolution as ours, each coastal system must be represented by at least two spaces (SLS and USS) and pelagic marine systems by at least three spaces (SLS, WCS and USS). In the following we represent functional segments with the corresponding letters and in square brackets behind them the spaces in which the associated processes end or take place (syntax example: A [WCS], OC position change ending in the WCS).

Based on the unique combinations of 1) the sequence of functional segments and 2) the involved spaces, we now define three *closed remineralisation* and two *rDOC loops* (Figure 1 and Table 2).

We the remineralisation loops comprise: a *surface layer remineralisation loop (SLRL)*, a *water column remineralisation loop (WCRL)*, and an *upper sediment remineralisation loop (USRL)* (Table 2). All three loops include pathways on which OC is remineralised to DIC (D), which is taken up by primary producers in the SLS (F [SLS]). The functional segments 'OC

position change' (A) and 'DIC upward position change' (E) as well as the space in which the OC is remineralised distinguish the remineralisation loops. The WCRL includes pathways that lead to a downward position change of OC into the WCS, remineralisation in the WCS, and an upward position change of DIC into the SLS, where it is taken up (WCRL: A [WCS] → D [WCS] → E [SLS] → F [SLS]). An exemplary WCRL pathway involves OC uptake by zooplankton in the SLS, its migration into and respiration in the WCS, and the upward mixing of the resulting DIC into the SLS where it is taken up by primary

producers (WCRL: A [WCS] → D [WCS] → E [SLS] → F [SLS]). If zooplankton respiration occurs in the SLS, the pathway belongs to the SLRL (SLRL: D [SLS] → F [SLS]). We define the USRL analogous to the WCRL, but with remineralisation taking place in the USS (USRL: A [USS] → D [USS] → E [SLS] → F [SLS]).

The two functional segments 'Formation of rDOC' (B) and 'rDOC conversion to DOC' (C) in the SLS are part of another set of closed loops, the rDOC loops (Figure 1 and Table 2). The rDOC loops describe the change of labile OC to more recalcitrant forms, its persistence in the system, and its return to bioavailable forms in the SLS. We differentiate a *short rDOC loop*

*(SrDOCL)*, rDOC that accumulates in the surface waters on time scales of human life, and a *long-term rDOC loop (LrDOCL)*, rDOC that persists in the entire water column on geological time scales. The short-term rDOC loop is defined by the 'Formation of rDOC' (B) and 'rDOC conversion to DOC' (C) in the SLS (SrDOCL: B [SLS] → C [SLS]), while the rDOC long-term loop additionally comprises the functional segment 'OC position change' (A), with accumulation mostly or even entirely in the

WCS (LrDOCL: B [SLS] → A [WCS/USS] → A [SLS] → C [SLS] or A [WCS/USS] → B [WCS/USS] → A [SLS] → C [SLS]). In contrast to the remineralisation loops, we do not explicitly consider a rDOC loop in the upper sediment, as the temporal scales of rDOC produced there or in the water column overlap to our knowledge. Therefore, the long-term rDOC loop includes rDOC production in the USS alongside its transport to the WCS. It has to be noted, that only rDOC that reaches the surface and is converted back into more bioavailable forms in the SLS belongs to the LrDOCL (LrDOCL: ... A [SLS] → C

[SLS]). rDOC can for instance be part of the WCRL when remineralised in WCS (WCRL: B [SLS] → A [WCS] → D [WCS]

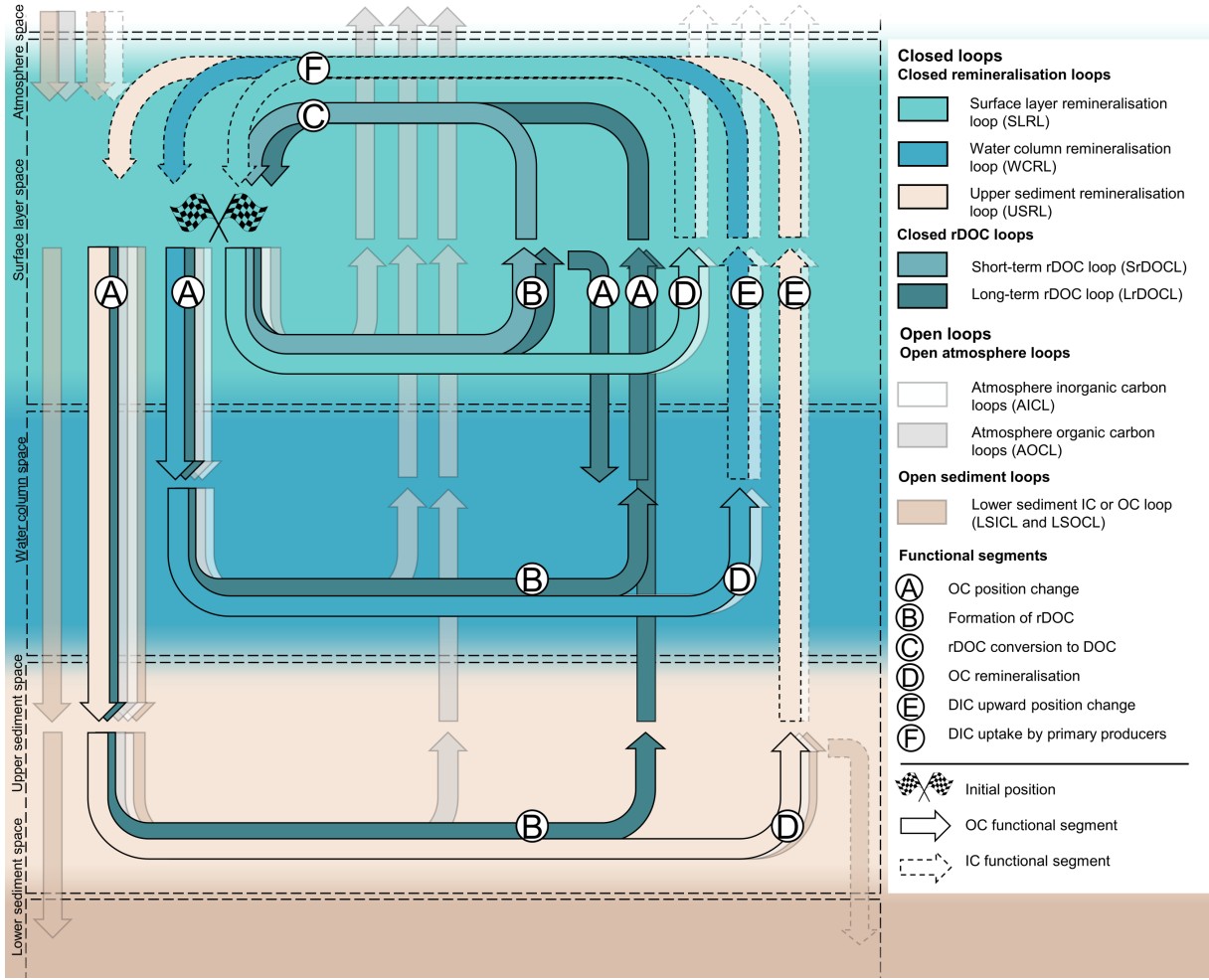

**Figure 1.** General structures of the marine OC cycling with three closed remineralisation and two closed rDOC loops, the spaces and the involved functional segments. 'Open' loops are only displayed with transparent colours as they are not our focus.

$\rightarrow$ E [SLS] $\rightarrow$ F [SLS]). Because of the climatic importance of rDOC, we distinguish rDOC from DOC as described before. Technically, however, rDOC represents a "storage" intermediate step of remineralisation or open loops.

All loops comprise a continuum of processes that are not addressed in the defined sequences of functional segments. For example, the SLRL also includes pathways on which OC is transported and processed below the SLS but returns to the SLS as OC to be remineralised and used by primary producers (SLRL: A [WCS] $\rightarrow$ A [SLS] $\rightarrow$ D [SLS] $\rightarrow$ F [SLS]). To avoid double counting when assigning pathways like this to one of our defined loops, two separation rules apply. The first rule states that the space of the ultimate remineralisation before entry and reuse in the SLS defines the remineralisation loop. OC that is remineralised several times in different spaces is part of the SLRL if it is last remineralised in the SLS before uptake by primary producers in the SLS. Similarly, OC belongs to the WCRL or USRL if it is ultimately remineralised in the WCS or

USS. The second rule states that rDOC leaving the surface or produced below the SLS always belongs to the LrDOCL (Table 2).

For the minimal description of the remineralisation and rDOC loops, the sequences of the above-defined functional segments are sufficient and unambiguous. However, users of the concept can identify and combine other functional segments to define different higher-resolution structures.

**Table 2.** Summary of sequences of functional segments and spaces defining the remineralisation and rDOC loops. The separation rule comes to play, when assigning a pathway to one of the defined loops. Spaces in square brackets indicate the spaces where the processes happen or end. Bold spaces are the naming spaces of this loop. Non-bold spaces are intermediate or "walk-through" spaces. Loops: Surface layer remineralisation loop (SLRL), Water column remineralisation loop (WCRL), Upper sediment remineralisation loop (USRL), short and long-term rDOC loop (SrDOCL, LrDOCL). Spaces: Surface layer space (SLS), Water column space (WCS) and Upper sediment space (USS). functional segments: OC position change (A), Formation of rDOC (B), rDOC conversion to DOC (C), OC remineralisation (D), DIC upward position change (E) and DIC uptake by primary producers (F).

| Closed Loops | Sequence of functional segments plus spaces | Separation rule |
|---|---|---|
| SLRL | D [**SLS**] $\rightarrow$ F [**SLS**] | Ultimate remineralisation in SLS before F |
| WCRL | A [**WCS**] $\rightarrow$ D [**WCS**] $\rightarrow$ E [SLS] $\rightarrow$ F [SLS] | Ultimate remineralisation in WCS before E and F |
| USRL | A [**USS**] $\rightarrow$ D [**USS**] $\rightarrow$ E [SLS] $\rightarrow$ F [SLS] | Ultimate remineralisation in USS before E and F |
| SrDOCL | B [**SLS**] $\rightarrow$ C [**SLS**] | Formation of rDOC in SLS and no A |
| LrDOCL | B [SLS] $\rightarrow$ A [**WCS/USS**] $\rightarrow$ A [SLS] $\rightarrow$ C [SLS] or A [**WCS/USS**] $\rightarrow$ B [**WCS/USS**] $\rightarrow$ A [SLS] $\rightarrow$ C [SLS] | Formation of rDOC in SLS with A or Formation of rDOC in WCS or USS |

Although we focus on the closed loops, it is noteworthy that there are parallel 'open' loops of carbon that close outside the marine systems, e.g. in the atmosphere. We define four structures of 'open' loops. The *atmosphere IC loops (AICLs)* describe the outgassing of DIC, produced in different spaces, to the atmosphere. The *atmospheric OC loops (AOCLs)* comprise the exit of marine OC, marine aerosols, volatile organic compounds (VOCs), and $CH_4$ through the surface to AS, e.g. via fish predation by birds or outgassing. The *lower sediment IC (LSOCL)* and *lower sediment OC loops (LSICL)* describe the burial 230 and lithification of carbon in the LSS, entering geological cycling.

### 3.2 Embedded processes, pools and agents

Having defined the structures of remineralisation and rDOC loops, we now add and describe global processes, pools and agents embedded in each functional segment (Figure 2 and Table 3). This addition allows to define structures with higher resolution and to link and complement our concept with existing ones. Global in this context means that the process mechanisms are 235 globally valid, but that the frequency, extent, initialisation and triggers of these processes differ. We focus on non-anthropogenic processes and the previously defined functional segments. This means that, for example, upward position changes of POC or DOC are not resolved.

Two of the three remineralisation loops include the functional segments OC position change (A) and DIC upward position change (E). Processes belonging to functional segments A and E include sinking, diffusion and advection, and direct and indirect biota-induced transport.

Organic compounds that sink from one space in the water column to another are usually either large or dense, or escape consumption or dissolution in the upper space (De La Rocha, 2006). Sedimentation and compaction by subsequent matter is the analogous process within the sediment-water interface and sediment. Matter is compacted by the weight deposited over it and "sinks" as it loses volume. Sinking and sedimentation always act downwards and are confined to POC. Gravity-induced sinking (and sedimentation) is thus part of any functional segment A of POC (Figure 2).

(r)DOC (DOC and rDOC) and DIC potentially diffuse in all directions, following large- or small-scale gradients in the water column, at the water-sediment interface and in the pore water of the sediment. We assume that (r)DOC concentrations decrease with depth (Hansell, 2013) but are higher in the sediment than in the overlying water (Burdige et al., 1999; Rowe and Deming, 2011) and that DIC concentrations increase with depth (Oka, 2020). Following these gradients, (r)DOC diffuses downwards in the water column and upwards in and out of the sediment (A of (r)DOC in Figure 2) and DIC always diffuses upwards (E). The upward diffusion of non-refractory DOC from the sediment is not considered in the defined functional segments as upward movements are not common to all pathways of the remineralisation loops.

Other physically induced position changes are related to water or sediment mass movements based on advection. These include large-scale upwelling and downwelling water movements, seasonal mixing, wind-induced turbulence and eddies, and storm-induced resuspension. Advection is globally applicable although its direction, magnitude, and frequency vary. The advection-induced position change occurs in all functional segments A and E. Advection does not act downwards into the sediment but upwards in the form of resuspension. Resuspension is only included for rDOC and is limited to the upper part of the sediment, as physical perturbation do not commonly reach below 10 cm (Boudreau, 1998; Bunke et al., 2019).

Biota-induced transport involves the direct transport of OC in the living tissue of migrating organisms (e.g. a fish feeds in the SLS, migrates down, and dies in the WCS) as well as the internal flux of OC in organisms that span different spaces (e.g. macrophytes living in the SLS and the USS (Middelburg, 2019)). Organisms change their position in the water column (e.g. via diel vertical migration (Steinberg et al., 2002)) or in the sediment (e.g. via burrowing (Middelburg, 2019)) and produce faecal pellets or die after the position change. The result of direct biota-induced position change is POC of all sizes, e.g. living organisms and roots, faeces, and carcasses. Direct biota-induced position change works in all directions and is involved in all functional segments A of POC.

Indirect biota-induced transport comprises biogenic turbulence (Kunze et al., 2006; Huntley and Zhou, 2004), and induced drift, which describes the transport of substances that adhere to the bodies of swimming organisms (Katija and Dabiri, 2009). Indirect biota-induced position change in the sediment is related to among others bioturbation (Berke, 2010), associated with sediment reworking and resuspension, and bioirrigation (Kristensen et al., 2012), which leads to inflows of ocean water into the sediment. Indirect biota-induced position change works in all directions and is involved in all functional segments of A for (r)DOC and POC and E in the water column and the sediment.

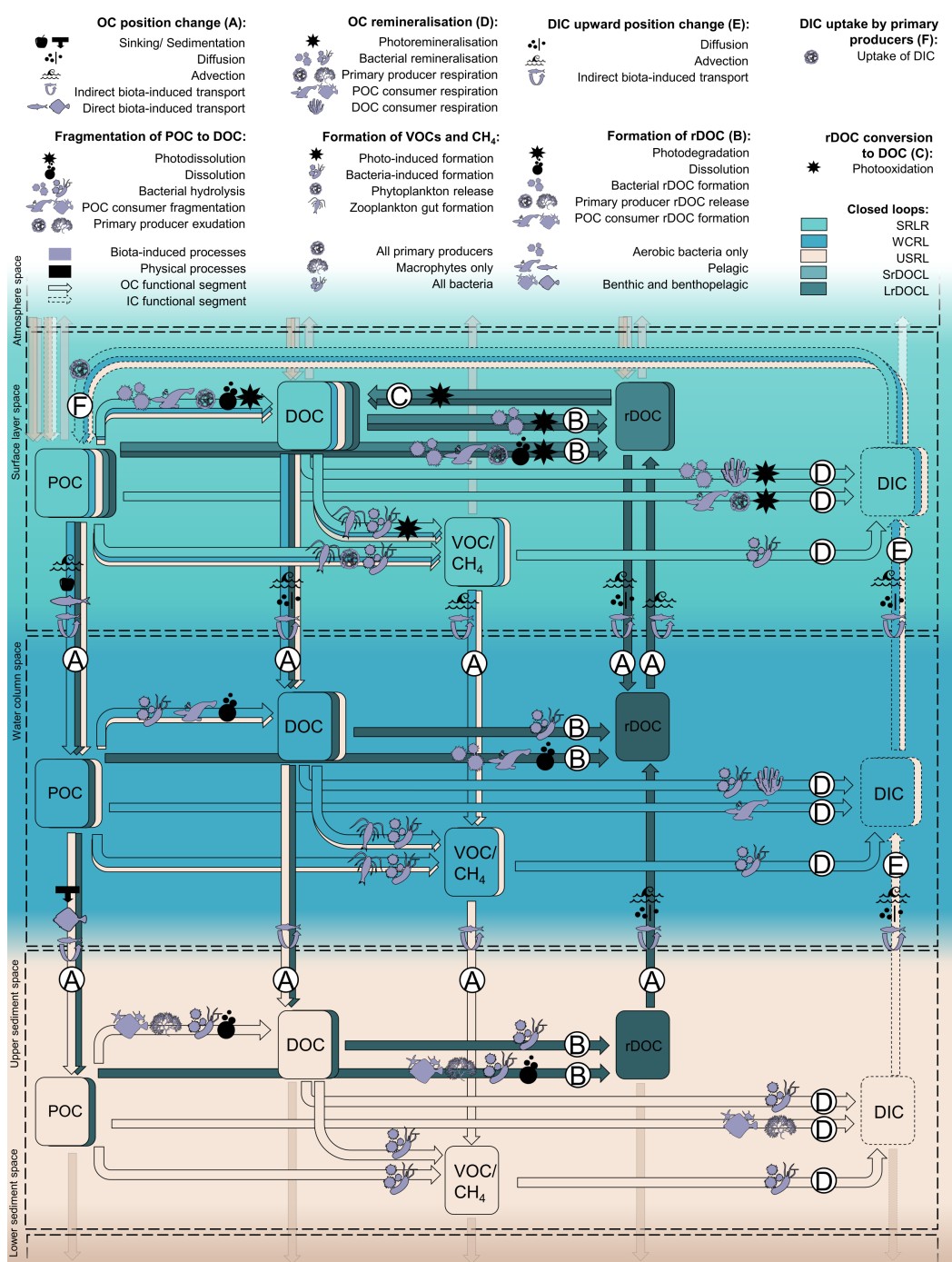

**Figure 2.** Defined OC structures with functional segments A-F, spaces and embedded processes, OC pools and involved organisms. 'Open' loops are indicated by transparent colours. Organisms can be agents (producing DOC by sloppy feeding) and part of the carbon pool (consumers as part of POC respire DIC) at the same time. Fragmentation processes and pathways for DOC and VOCs/ CH$_4$ are included. As they are not resolved in the loops we defined, these pathways are not marked with capital letters.

The following processes belong to the functional segment OC remineralisation (D). We define remineralisation as the provision of DIC based on OC and restrict it to the spaces above the LSS, assuming that remineralisation in the LSS is negligible.

Light-induced photoremineralisation, the only physically induced remineralisation, directly oxidises DOC and POC to IC (Mopper and Kieber, 2002; Mayer et al., 2009) and works only in the SLS. We include this process in D in the SLS.

Bacteria and archaea remineralise DOC in functional segment D in every space above LSS, also under different oxygen conditions. The DOC is either of allochthonous origin (e.g. entering via riverine input (Dai et al., 2012)), or of autochthonous origin based on living or non-living POC. For instance, POC dissolves while sinking (Carlson and Hansell, 2015), is fragmented by turbulence (Ruiz, 1997; Briggs et al., 2020), or photodissolved (Mayer et al., 2006). Consumers reduce the size of organic POC by sloppy feeding on living and non-living POC by e.g. zooplankton coprorhexy (Lampitt et al., 1990), by producing small metabolites, by excreting DOC (Lampert, 1978) or by swimming or moving (Dilling and Alldredge, 2000; Goldthwait et al., 2004). Further, primary producers exudate DOC in the water column (e.g. under nutrient-limited conditions or viral lysis (Azam and Malfatti, 2007)) and in the sediment (by macrophytes (Duarte and Cebrián, 1996)). Bacteria, for their part, hydrolyse POC to DOC (Smith et al., 1992) and additionally release DOC by viral lysis (Middelboe et al., 1996).

The transformation from POC to DOC (arrows from POC to DOC, Figure 2) that takes place before bacterial remineralisation are not defined as functional segment of the remineralisation loops, as not every OC compound needs to undergo one of these changes to be remineralised. However, when e.g. considering only DOC-based pathways, the change in OC size from POC to DOC can be defined as a common functional segment and used to define structures such as POC-DOC remineralisation loops.

In addition, bacteria can oxidise VOCs and CH4 as e.g. shown in Halsey et al. (2017) (D of VOCs/ $CH_4$ in Figure 1). The VOCs and CH4 origin from abiotic processes such as photochemical degradation of DOC (Kieber et al., 1989) and biogenic processes, e.g. production by phytoplankton (Lenhart et al., 2016) and zooplankton in anaerobic areas of their guts (Weber et al., 2019; Schmale et al., 2018).

Another form of remineralisation is respiration by living organisms other than bacteria. Primary producers respire in the photic SLS. The roots of macrophytes additionally produce DIC in the USS at night (Pedersen et al., 1995). Higher trophic levels, POC consumers (e.g. zooplankton and fish) and non-bacterial DOC consumers (e.g. suspension-feeding sponges at the sediment-water interface (Wooster et al., 2019)), also remineralise by respiration. Therefore, we include remineralisation by primary producers in functional segments D in the SLS and USS, respiration by DOC consumers in the SLS and WCS, and respiration by POC consumers in all spaces with aerobic conditions above the LSS.

Once OC is remineralised to DIC, this DIC is transported by the above-described processes of position change to the SLS (E [SLS]). Subsequently, primary producers take up the DIC for photosynthesis (F [SLS]) and close the remineralisation loops.

The rDOC loops include the formation of rDOC (B), the reconversion to DOC in the SLS (C), and, in case of the long-term loop, position change of OC (A). We present some of the involved abiotic and biotic processes, which have been reviewed e.g. in Legendre et al. (2015).

**Table 3.** Processes embedded in the functional segments of the defined loops. Italic pools are products of the processes. Processes end or take place in the spaces in square brackets in the loop syntax.

**OC position change (A)**

| Process | Loop syntax | Process description | Pools | Organisms | Directions |
|---|---|---|---|---|---|
| **Sinking** | WCRL: A [WCS] LrDOCL: A [WCS] | Gravitational sinking | POC | | Downwards |
| **Sedimentation** | USRL: A [USS] LrDOCL: A [USS] | Sedimentation of sinking matter | POC | | Downwards |
| **Diffusion** | WCRL: A [WCS] | Diffusion in the water column and pore waters | DOC | | Downwards |
| | LrDOCL: A [WCS] | | rDOC | | Downwards, upwards |
| **Advection** | WCRL: A [WCS] | Up- and downwelling, mixing, turbulence, eddies | POC, DOC, VOCs, CH$_4$ | | Downwards |
| | LrDOCL: A [WCS] | | POC, DOC, rDOC | | Downwards, upwards |
| **Indirect biota-induced transport** | WCRL: A [WCS] USRL: A [USS] | Biota-induced turbulence, induced drift, digging, burrowing, bioirrigation, sediment reworking | POC, DOC, VOCs, CH$_4$ | Swimming and moving species (pelagic, bentho-pelagic and benthic) | Downwards |
| | LrDOCL: A [WCS] LrDOCL: A [USS] | | POC, DOC, rDOC | | Downwards, upwards |
| **Direct biota-induced transport** | WCRL: A [WCS] USRL: A [USS] | Transport in living tissue or OC distribution in organisms spanning several spaces | POC | Swimming and moving species (pelagic, bentho-pelagic and benthic), organisms spanning several spaces (e.g. kelp) | Downwards |
| | LrDOCL: A [WCS] LrDOCL: A [USS] | | | | Downwards, upwards |

**Formation of rDOC (B)**

| Process | Loop syntax | Process description | Pools | Organisms | Directions |
|---|---|---|---|---|---|
| **Photo-degradation** | SrDOCL: B [SLS] LrDOCL: B [SLS] | Degradation of labile to recalcitrant OC by UV light | DOC, POC, *rDOC* | | |
| **Dissolution** | SrDOCL: B [SLS] LrDOCL: B [SLS] LrDOCL: B [WCS] LrDOCL: B [USS] | Dissolution due to sinking (enhanced by bacteria) or pore-water interactions | POC, *rDOC* | | |

| Bacterial rDOC formation | SrDOCL: B [SLS] <br> LrDOCL: B [SLS] <br> LrDOCL: B [WCS] <br> LrDOCL: B [USS] | Release of capsular material and rDOC under e.g. stress conditions | DOC, POC, *rDOC* | Bacteria, viruses | |
|---|---|---|---|---|---|
| Primary producer rDOC release | SrDOCL: B [SLS] <br> LrDOCL: B [SLS] <br> LrDOCL: B [WCS] <br> LrDOCL: B [USS] | Release of rDOC | POC, *rDOC* | Phytoplankton and e.g. macrophytes | |
| POC consumer rDOC formation | SrDOCL: B [SLS] <br> LrDOCL: B [SLS] <br> LrDOCL: B [WCS] <br> LrDOCL: B [USS] | Direct (excretion) or indirect release (e.g. via sloppy feeding) of rDOC | POC, *rDOC* | POC consumers (pelagic, bentho-pelagic and benthic) | |

**Conversion of rDOC to DOC (C)**

| Process | Loop syntax | Process description | Pools | Organisms | Directions |
|---|---|---|---|---|---|
| Photooxidation | SrDOCL: C [SLS] <br> LrDOCL: C [SLS] | Photochemical conversion rDOC to DOC | rDOC, *DOC* | | |

**OC remineralisation (D)**

| Process | Loop syntax | Process description | Pools | Organisms | Directions |
|---|---|---|---|---|---|
| Photo-remineralisation | SLRL: D [SLS] | Direct UV remineralisation | POC, DOC, *DIC* | | |
| Bacterial remineralisation | SLRL: D [SLS] <br> WCRL: D [WCS] <br> USRL: D [USS] | Bacterial DOC (VOCs)-based respiration | DOC, VOCs, $CH_4$, *DIC* | Bacteria and archaea | |
| Primary producer respiration | SLRL: D [SLS] <br> USRL: D [USS] | Respiration of primary producers | POC, *DIC* | Phytoplankton and e.g. macrophytes | |
| POC consumer respiration | SLRL: D [SLS] <br> WCRL: D [WCS] <br> USRL: D [USS] | Respiration of POC consumers | POC, *DIC* | POC consumers (pelagic, bentho-pelagic and benthic) | |
| DOC consumer respiration | SLRL: D [SLS] <br> WCRL: D [WCS] | Respiration of DOC consumers | DOC, *DIC* | DOC consumers (filter feeders) excluding bacteria | |

**DIC upward position change (E)**

| Process | Loop syntax | Process description | Pools | Organisms | Directions |
|---|---|---|---|---|---|
| Diffusion | WCRL: E [SLS] <br> USRL: E [SLS] | Diffusion in the water column and pore waters | DIC | | Upwards |

| Advection | WCRL: E [SLS] USRL: E [SLS] | Down- and upwelling, mixing, turbulence and eddies, physical induced resuspension | DIC | | Upwards |
| Indirect biota-induced transport | WCRL: E [SLS] USRL: E [SLS] | Biota-induced turbulence, induced drift, digging, burrowing, bioirrigation, sediment reworking and related processes | DIC | Swimming and moving species (pelagic, bentho-pelagic and benthic) | Upwards |

**DIC uptake by primary producers (F)**

| Process | Loop syntax | Process description | Pools | Organisms | Directions |
|---|---|---|---|---|---|
| Uptake of DIC | SLRL: F [SLS] WCRL: F [SLS] USRL: F [SLS] | Photosynthesis | DIC, *POC* | Phytoplankton and e.g. macrophytes | |

UV light can change the lability and increase recalcitrant components of the DOC pool via photodegradation (Benner and Biddanda, 1998; Hansell, 2013)(B [SLS]). Biota supply rDOC via successive microbial processing of DOC (Jiao et al., 2010, 2011), the release of capsular material by bacteria (Stoderegger and Herndl, 1998), bacterial hydrolysis of POC (Jiao
et al., 2011), bacterial stress responses to low-labile DOC and unfortunate nutrient conditions (Stoderegger and Herndl, 1998), and successive consumption by higher trophic levels (Jiao et al., 2011). In addition, some phytoplankton directly exudates rDOC (Jiao et al., 2011). Both microbes and phytoplankton also release rDOC due to viral lysis of host cells (Jiao et al., 2011). Furthermore, processes that convert living and non-living POC into DOC, e.g. dissolution, can dilute DOC to the point where it can no longer serve as sufficient nutrition for microbes and can be considered technically recalcitrant (Arrieta et al., 2015)
(Figure 2, arrow from POC to rDOC).

rDOC that stays in or returns to the SLS, via the position change processes described above (A [SLS]), can be converted back to more bioavailable forms by photooxidation (C [SLS]) (Kieber et al., 1989). We consider pathways with other rDOC removal processes, such as direct light-induced oxidation from rDOC to DIC (Shen and Benner, 2018), sorption of rDOC into POC (Hansell et al., 2009) and hydrothermal removal mechanisms in hydrothermal vents or the Earth's crust (Lang et al.,
2006), as parts of the closed remineralisation or 'open' loops. Once the rDOC is converted to DOC in the SLS, the rDOC loops are closed.

Based on these embedded processes, pools, and agents, we can now define structures of higher resolution. For example, for SLRL, six structures can be defined based on the carbon pools involved: a POC-SLRL, a POC-DOC-SLRL, a POC-DOC-VOC/CH$_4$-SLRL and a POC-VOC/CH$_4$-SLRL, as well as a DOC-SLRL and a DOC-VOC/CH$_4$-SLRL. Depending on the
research question or desired level of detail, multiple structures can be defined based on the processes and agents involved. The higher the resolution of the structure, the more the structures resemble descriptions of individual pathways. In the following

discussion, we use the example of the biological carbon pump to show how different structures can look like and which insights e.g. a comparison of such different structures can provide.

## 4    Discussion

Our concept of the marine OC cycle condenses pathways to superordinate structures and provides an overview of embedded processes, pools, and agents, which allows resolving structures of smaller scale and higher resolution. Our overarching structures complement existing concepts of OC pathways and processes in the ocean, providing a basis for using a consistent terminology. As such, the concept facilitates comparing different definitions of conceptualised pathways, integrating new findings and placing, for example, pathways of finite length scale in a broader framework.

To discuss some of these aspects in an application example, we translate pathways of the biological carbon pump (BCP) into our concept (Figure 3). Based on the definition of Giering and Humphreys (2020)[1], who define the BCP as "the collection of marine biogeochemical processes that convert dissolved inorganic matter in the surface ocean into biomass and transport this to the ocean interior, where the biomass is returned to its original dissolved inorganic forms", we illustrate different structures with different resolutions and choices of pathways.

Using the syntax of our concept and functional segments A-F, the defined BCP involves the uptake of inorganic carbon into biomass in the surface waters (F [SLS]) and the OC position change to the interior of the ocean (A [Ocean Interior]), where it is remineralised to DIC (D [Ocean interior]) (Figure 3 panel (a)). For simplicity, we disregard rDOC, VOCs and $CH_4$ and start again with the previously introduced initial position. As it is not clarified in the definition, we assume that the ocean interior does not contain the USS and define it as WCS. Based on this restriction of "ocean interior" we classify the BCP as part of the

WCRL or the corresponding atmospheric inorganic carbon cycle (AICL). Note that we need to add functional segment E to count the BCP to the WCRL as E is not included in the defined BCP.

If we now resolve the OC pools involved in the BCP (here POC and DOC), we can define three BCP structures of higher resolution (panel b). Each of these structures defines a part of the BCP. All together, they capture all pathways of the defined BCP. Taking only pathways with a specific set of processes into account, produces structures that do no longer comprise all

pathways of the BCP. For example, focusing on pathways with direct biota-induced transport (A), results in seven structures. These structures only serve structure 2 from panel (b) (Figure 3) and thus represent only a part of the defined BCP. This part resembles other concepts of BCPs such as the mesopelagic-migrant pump and the seasonal lipid pump (Boyd et al., 2019). Focusing on the purely physically induced pathways of the BCP leads to six different structures that do not resolve remineralisation (D) and DIC uptake (F) as they are non-physical processes (Figure 3, panel (d)). These six structures could

potentially serve all of the higher-level structures in panel (b) of the defined BCP if we add D and F, but do also not cover all the pathways of the BCP (see panel (c)). Nevertheless, the structures in d) resemble some, often more traditional, concepts of the BCP, e.g. from Hansell and Carlson (2001) and De La Rocha (2006), which do not explicitly consider DIC uptake and remineralisation.

---

[1]If not mentioned differently, we always refer to the BCP definition by Giering and Humphreys (2020) in the following discussion.

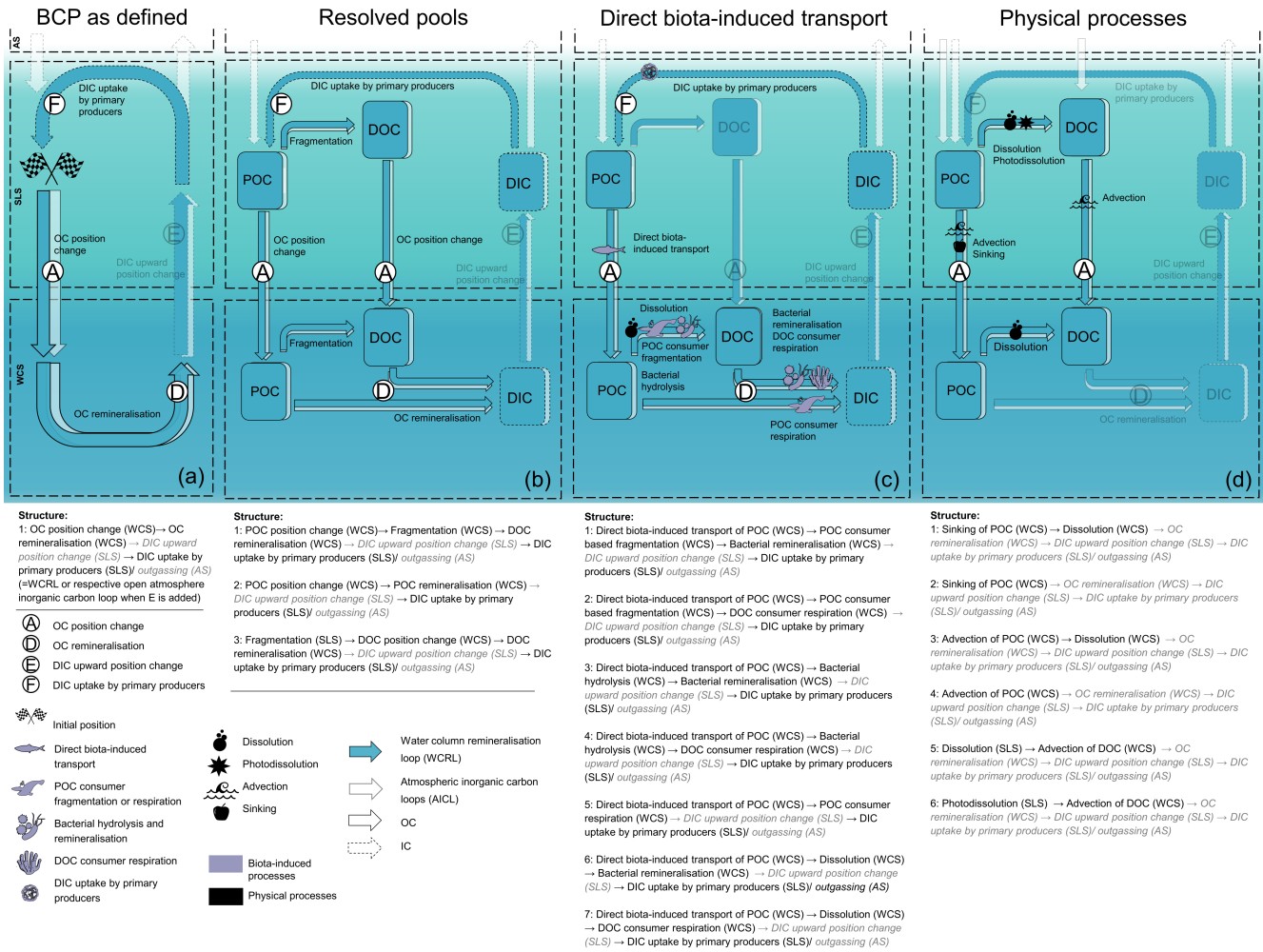

**Figure 3.** The translation of the BCP defined by Giering and Humphreys (2020) into our concept. Panel (a) shows the superordinate BCP structures based on the definition. Panel (b) additionally resolves involved pools. Panel (c) and (d) resolve choices of processes: (c) pathways with direct biota-induced transport and (d) pathways with only physical processes. Transparent or italic/grey functional segments are not explicitly included in the definition of the BCP or the selected pathways. Note that all structures are part of the WCRL or AICL when adding functional segment E to a) - c) and D, E and F to d). Note further, that panel d) does only belong to the defined BCP if functional segments D and F are added.

Integrating the BCP definition of Giering and Humphreys (2020) into our concept illustrates where the BCP concept shows

ambiguity and may need to be refined to concretise which pathways belong to the BCP and which do not. Giering and Humphreys (2020) give "ocean interior" as the spatial constraints of the pathways of the BCP. We translate "ocean interior" as WCS, as no further depth constraints are given. Other BCP definitions constrain depth more concretely, for example describing the BCP as export (Buchan et al., 2014; Hansell and Carlson, 2001) and sequestration fluxes (Sigman and Haug, 2004) acting

at depths below 100 m and 1000 m (Passow and Carlson, 2012). This ambiguity of the space in which the BCP operates means

that we may identify pathways as part of the BCP that are not, while excluding others that actually are. It is therefore necessary to define the spaces in which the BCP operates more distinctly. Subdividing the WCS into several spaces, e.g. a space below a sequestration depth, may thus be more appropriate for the representation of the BCP, as the definition of spaces allows a refinement of the pathways that belong to a structure.

A similar vagueness as with the spaces applies to the OC pools involved. Does the BCP include pathways based on DOC in

the SLS, e.g. as defined by Honjo et al. (2014), or not as defined by De La Rocha (2006)? DOC is one of the relevant carbon fluxes to the deep sea, especially in oligotrophic areas (Roshan and DeVries, 2017). Therefore, and because the definition of Giering and Humphreys (2020) does not explicitly exclude the DOC pool, we add DOC to our illustration of the BCP in panel (b). However, the presentation would also work without DOC. In such a case, our concept shows which pathways are missing by dispensing DOC.

Illustrating what is missing also allows placing individual pathways and concepts such as the BCP in a broader framework. For example, mentioning pathway section E is a must to place the BCP in the OC cycle, as there is no dead end in nature. Furthermore, our approach helps to identify how different sub-concepts fit into more general definitions (panels (b)-(c) compared to panel (a)), but also where some inconsistencies might occur, e.g. remineralisation included a)- c) or not d). In addition, it facilitates identifying which pathways are not resolved and the potential informative value of studies based on a limited

number of pathways. In panel (c), for example, the DOC in the WCS comes only from fragmentation of POC. If fragmentation processes decrease significantly, this does not necessarily mean a decrease in remineralisation of DOC (D [WCS]), as DOC can also originate from the SLS. A study based on the pathways as in panel (c) does not consider DOC from the SLS and therefore has limited informative value about changes in remineralisation of DOC. All mentioned considerations are already part of most studies and publications. But we provide a new tool to systematise these considerations and make them more comparable.

The BCP example also shows how new concepts and processes can be integrated into our concept. Panel (d) resembles more traditional definitions of the BCP, which focus mainly on physically driven processes. The role of organisms, particularly higher trophic levels, was considered quantitatively secondary and therefore neglected. Now, however, the contribution of this biota is recognised as relevant to the carbon cycle. For example, large migratory species link to nutrient distribution and overall mixing (Roman and McCarthy, 2010), zooplankton have a significant influence on carbon export (Steinberg and Landry, 2017),

reptile falls provides an alternative carbon pathway to the sediment (McClain et al., 2019), and fish and mammals contribute to the marine OC cycle through various processes (Martin et al., 2021). With these processes, many new structures emerge, some of which we resolve in panel (c). Our concept provides overarching structures that users can bring to life to integrate new insights. Processes, organisms, pathways, and loops can easily be added, changed, or deleted to incorporate new findings or specific systems into general structures.

By generalizing structures and providing a congruent visual representation, our concept may reduce the potential for misunderstandings of the marine OC cycle potentially and unintentionally caused by visual concepts of finite length scale. An example of such a potential misunderstanding is the representation of pathways transporting DIC to depths without resolving what happens to the DIC after in some earlier visual OC concepts (as e.g. discussed for Boscolo-Galazzo et al. (2018) in the

introduction). While these representations are justified for finite length scale studies, this visual decoupling can lead to the false impression that increased transport of OC to the deep ocean always leads to increased sequestration and storage of atmospheric carbon. However, increased OC export is not necessarily accompanied by increased carbon storage, which depends, among other things, on the ratio of regenerated to pre-formed nutrients and on the carbon that escapes from the deep ocean (Gnanadesikan and Marinov, 2008). The export of carbon to the deep sea is part of carbon processing, but not the whole story, as we can also see from the example of the definition of the BCP. We propose to use a concept like ours as a reference concept to address the increasingly interdisciplinary scientific community, to strengthen the coherence of (visual) concepts and to identify the overarching structures of individual pathways.

The provision of overarching structures comes at the cost of not capturing the complexity of the marine OC cycle. Each OC compound travels its pathway through the OC cycle. An OC compound in the surface ocean may end up on the surface or in the deep sea, be decomposed, or become recalcitrant, to name just a few possibilities. Each pathway is unique in its sequence of processes. So, there is a multitude of possible pathways. An all-encompassing description of these possibilities is, therefore, neither possible nor meaningful. Accordingly, our concept does not want to and cannot resolve individual pathways. On the contrary, it focuses on overarching structures and the minimal functional segments necessary to describe them. Hence, our concept reduces many pathways to a sequence that does not capture their full extent, biological relevance, complexity, and temporal dimension.

Moreover, our concept is purely abstract and not capable of quantification or forecasting expected changes. It is a skeleton that needs to be filled with life. Further, it proved difficult to find an unambiguous language and visualisation for the concept. For example, we depict organisms that are a pool and organisms that are agents with the same symbol. Adjustments of terms and symbols appear reasonable as soon as users identify problems. We hope that the concept will grow, improve and become more complete with use.

## 5 Conclusion

We propose a general (visual) concept for the marine part of the organic carbon cycle. It complements and integrates existing concepts and defines overarching OC structures such as remineralisation and rDOC loops and the processes, pools and agents involved. Building on concepts that focus on individual or a subset of marine OC pathways, our concept identifies general structures of all pathways. Details and complexity are disregarded in favour of a systematic structures that can facilitate the identification and comparison of concepts, pathways, pools and studies. The concept can be adapted to a wide range of questions, pathway choices, resolutions and thus serve as a basis for discussion and reference to understand the current and future marine OC dynamics arising from the multiplicity of OC pathways and the human influence on them.

*Data availability.* The literature-based pathway concept is attached as supplement A.

*Author contributions.* M.I.E.S. designed the study, conducted the research and prepared the manuscript. I.H. designed the study and contributed to the manuscript.

*Competing interests.* The authors declare that they have no conflict of interest.

*Funding.* This study has been funded by the Deutsche Forschungsgemeinschaft (DFG, German Research Foundation) under Germany's Excellence Strategy- EXC 2037 'Climate, Climatic Change, and Society'- Project Number: 390683824, contribution to the Center of Earth System Research and Sustainability (CEN) of Universität Hamburg.

*Acknowledgements.* Our special thanks go to Laurin Steidle, Alex Kitts, Felix Pellerin, Rémy Asselot, Isabell Hochfeld, Josefine Herrford and Jana Hinners for their valuable feedback and Scott Dorssers for his support in finalising the literature-based pathway concept. We thank one anonymous referee and Gwenaelle Gremion for their helpful criticism and comments.

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
