# Peer review of "There and back again, a journey of many pathways: conceptualising the marine organic carbon cycle"

_Ocean Science, 2021_

## Author Response (AR1)

**List of changes:**

- Changed the title and subtitles of section 2, 3.1 and 3.2.

- Changed the wording from 'conceptual model' to 'concept' as the term 'conceptual model' proved difficult as highlighted by both referees.

- Rewrote the introduction following the comments of referee 1 and 2 that the motivation of our work needed to be stronger.

- Rewrote the discussion. Now the discussion is built around an application example. Using the suggestion of referee 1, we used the concept of the biological carbon pump for this.

- Added new figure 3 for the discussion.

- Added flow syntax as suggested by referee 1.

- Added an atmosphere space as suggested by referee 2.

- Changed name of the surface remineralisation loop to surface layer remineralisation loop (SLRL) as suggested by referee 1.

- Modified table 1, table 2, table 3, figure 1 and figure 2 to ensure matching terms, complete legends and to add new spaces, names of loops and the flow syntax.

- Revised terms to ensure that all terms are used in the same way. Removed terms that seem to be misleading, e.g. explicit, and added more insightful terms, e.g. superordinate, higher-level and subordinate.

- Deleted unnecessary references to tables and figures as suggested by referee 2.

- Changed the order of section 2 to ensure a better reading and information flow.

- Modified the lagoon analogy in section 3.1 answering comments of referee 2.

- Added a methodological concept as supplement B for better understanding of the conceptualisation process.

- Rewrote the abstract as the introduction and discussion changed drastically.

- Changed all words to UK spelling.

**Changes made to comments of referee 1:**

Overall, while I am intrigued by the manuscript, and can envisage how its insights might be used, I cannot currently recommend it for publication. By being presented in a highly abstracted way, and by avoiding specific examples of its use, it feels as if any potentially interested users still have a mountain to climb. I do understand why the authors have made these choices for the manuscript. However, I suspect most readers will find the concept of interest, but will be dissuaded from pursuing it because it is unclear what next steps are needed to make best use of it.

**We added an application example in the discussion (p.17 f).**

In terms of its revision on this point, what I think would help it be more clearly relevant is its use in an example. From my own area – global-scale marine biogeochemical modelling – I could see it potentially providing a framework for identifying and, more importantly, *quantifying* major carbon flows between different models and, potentially, observations. In this way, the details of individual models could be ignored to focus instead on major pathways, including how these change across simulations of the present and future. Adapting the manuscript to include a clear instance of the use of its conceptual framework feels necessary to me.

**We added an application example in the discussion (p.17 f).**

In terms of its revision on this point, what I think would help it be more clearly relevant is its Table 3's presentation of the conceptual model's "language" is necessary, but it seems very clunky (see comments below); I don't know how this can be avoided, however

**We added a pathway syntax and reduced content of table 3 (p.13-15).**

Abstract: the manuscript would do well to make clear somewhere that ocean uptake of anthro CO2 has little to do with the biological focus of this manuscript; the opening statement of the abstract is implying that the reverse is true

**We rewrote the abstract.**

Abstract: recalcitrant DOC is poorly defined at this point; this might not matter for the framework here, but the authors should note the work of Arrieta et al. (2015; https://doi.org/10.1126/science.1258955) on the bioavailability of low concentrations of DOC molecules

**We refrain from adding complexity to the abstract, as we highlight the different concepts of what rDOC is at a later stage (e.g. p. 3 L105 f).**

Abstract: regarding the absence of applications in this manuscript, one use this qualitative approach could be put to is creating a systematic structure of the ocean's carbon cycle that quantitative datasets or models can be aligned with; that would simplify both obs and models to allow simple comparisons of reservoirs and fluxes

**We rewrote the abstract and used the feedback of referee 1 for the motivation in the introduction and the abstract.**

Abstract: the manuscript would do well to make clear somewhere that ocean uptake of anthro CO2 has little to do with the biological focus of this manuscript; the opening statement of the abstract is implying that the reverse is true

**We rewrote the abstract (p.1).**

Abstract: regarding the absence of applications in this manuscript, one use this qualitative approach could be put to is creating a systematic structure of the ocean's carbon cycle that quantitative datasets or models can be aligned with; that would simplify both obs and models to allow simple comparisons of reservoirs and fluxes

**We added this motivation to our reasoning chain in the introduction and discussion, e.g. p.3 L72ff.**

Pg. 1, ln. 16-18: this sentence is confusing; what does "it is the first concept" mean?; models of the marine system are not always structured around e.g. export; also, the span of processes here is already covered in some models; so it's unclear why this framework is special

**We deleted this part and modified our reasoning chain in the introduction.**

Pg. 1, ln. 18-19: point 1 is good; it's what I've identified above; point 2 is also good, although it doesn't introduce anything new on this point; models usually include all of these exit points because they have to; in Earth system models, for instance, the budgeting of C is critical, so these exits are explicitly there (and usually monitored diagnostically)

**We deleted this part and modified our reasoning chain in the introduction.**

Pg. 1, ln. 20-21: It *needs* to be quantitative if it's to help here; being qualitative and abstracted to basic processes will reduce its usefulness

**We deleted this part and modified our reasoning chain in the introduction.**

Pg. 2, ln. 28-29: But pathway options are already explicitly included in quantitative models; those move C between different reservoirs, and between locations in the case of 3D models; the conception and representation of the biological pump and its elevation of interior ocean CO2 is arguably already a "general pathway"

**We deleted this part and modified our reasoning chain in the introduction.**

Pg. 2, ln. 31: "inter alia" is a relatively uncommon latin expression; it might be better to replace with "among other things" for non-native English speakers

**Agreed. Changed it, among other things (p.2 L39, p.7 L177, p.20 L411) or among others (p.12 L270).**

Pg. 2, ln. 31-33: bit of bracket indiscipline here

**We revised the sentence reducing brackets (p.2 L38).**

Pg. 2, ln. 45: remineralisation is effectively the respiration of organic carbon to inorganic DIC by bacteria or other microbes; respiration more or less by definition turns organic C into inorganic CO2

**We deleted that sentence and made clear that we refer to inconsistent visualisation of respiration and its product (p.2 L50).**

Pg. 2, ln. 48-50: this sounds like the authors are trying to give a more holistic description of the marine carbon cycle, but it is unclear how this will be achieved

**We modified our reasoning chain in the introduction.**

Pg. 3, ln. 59-60: this is getting a bit opaque now; also, the export arrow is necessarily thinner at depth because there is simply less material the deeper one goes - this is a simple function of material being added only at the surface

**We deleted this part and modified our reasoning chain in the introduction (p.2 L45 f).**

Pg. 3, ln. 66-68: again, this sounds like it is merely making a case for a more holistic treatment of the C cycle, i.e. everything including the kitchen sink; noting that models, including conceptual ones, truncate the real world does not seem all that novel a point to make; and, as already noted, some models already go quite far towards including as much detail as is known

**We deleted this part and modified our reasoning chain in the introduction.**

Pg. 3, ln. 81: what about dissolved inorganic carbon?; it's only the largest ocean reservoir

**No changes made.**

Pg. 3, ln. 82: So ... straightaway some real-world detail is being dispensed with?; that sounds a little less than holistic

**No changes made.**

Pg. 3, ln. 84: you might want to be very clear on what you mean by "species" here; it's obviously not "biological species", but it also appears not to be "chemical species" either; it's somewhere in between

**We rephrased it to: The different foci and the limited spectrum of the pathways considered lead to concepts that complement each other through different resolutions (focusing on different processes or pools), but also promote partly overlapping sub-concepts. (p.2 L35-36).**

Pg. 3, ln. 85-85: any explanation for these arbitrary size limits?

**We changed it to: DOC smaller 0.2 µm and POC larger 0.2 µm (Kharbush et al. 2020) (p.4 L103-105).**

Pg. 3, ln. 88: again, rDOC may not be recalcitrant at all, merely at low concentration

**No changes made.**

Pg. 3, ln. 88: 1.5 to 40,000 years is quite a span; again - any explanation for these arbitrary timespan limits?

**We added some more information on the division provided by Hansell (2013) (p.4 L105f).**

Pg. 4, ln. 91-92: you could note that very few models go to the bother of subdividing DIC into its constituent species

**No changes made.**

Pg. 4, ln. 99: the use of "particle" here might be confusing; in a marine context, this could mean an actual particle of marine snow

**We will use the term OC compound for the parts in question (e.g. in section 2 and the first lines of section 3.1).**

Pg. 4, ln. 105-109: this is a big ask; if the conceptual model is not going to be quantitative, it has to do something special qualitatively to compensate; the introduction here has not made it clear what

**We revised our introduction and discussion to answer this comment.**

Pg. 4, ln. 108: "a delay of consumers"?; this is opaque; I think it would make more sense if "a delay of" was deleted; however, I may be misunderstanding what "delay" means here

**We changed it to spatio-temporal mismatch with consumers that favours sinking (p.4 L100).**

Pg. 4, ln. 111: "bases" -> "is based"

**Changed (p.4 L113).**

Pg. 4, ln. 112: "non-exclusive"?; this might need a clarifying remark - suggesting that a review was non-exclusive implies that it read *all* of the literature

**We changed it to: unsystematic literature review (p.4 L113).**

Pg. 4, ln. 115-118: OK, this sounds good so far; obviously I'm immediately wanting to assign numbers ...

**No changes made.**

Pg. 4, ln. 121: "under the given hygiene conditions"?; this is a strange qualifier to add without explanation; is this an oblique reference to the ongoing pandemic?; in any case, either explain or delete

**We deleted the sentence.**

Pg. 4, ln. 121: "example" -> "analogy"

**Changed (p. 6 L123 and L135, p.7 L161).**

Pg. 5, ln. 138: "position change" is an interesting one given that this can have radically different drivers; it can be physical, biological-gravitational, biological-migrational, etc., each of which can have distinct consequences; for instance, dissolved OC moved by physics will not interact with sediments in the same was as OC sinking gravitationally; is this a problem?; by having multiple OC routes out of the surface, the diagram would suggest possibly not, but then is opaque on what these different routes cover

**No changes made.**

Pg. 5, ln. 150: quite; per my previous point

**No changes made.**

Pg. 6, ln. 167: SRL -> SLRL; per SLS?

**We changed it to SLRL throughout the manuscript.**

Pg. 6, ln. 167-176: this makes sense, but one has to concentrate with all the acronyms; I've two suggestions here; 1. maybe parse out the examples with in-line "equations" of flow pathways; i.e. SLS -> WCS -> SLS; 2. it feels like you need a syntax for using all of these space, process, pathway interactions - yes, you can write sentences with them, but having a consistent way of writing them might make it easier to follow what a particular example is doing

**We added a flow syntax: Path segment [space] → Path segment [space] throughout the manuscript and in the figures and tables.**

Figure 1: as already noted, the significance ascribed here to rDOC formation is perhaps misplaced given that rDOC might only be recalcitrant because individual chemical constituents are at low concentration; the predominance of this process in this model seems disproportionate given that potentially more quantitatively important processes are downgraded and lumped into "OC remineralisation"

**No changes made.**

Pg. 14-16, Table 3: while this table tries to address a point I made earlier about allowing a consistent description of the pathways, the result seems very clunky to me; I can't immediately envisage a clear alternative, to be fair

**We adapted table 3 mostly following the comments of referee 2.**

Pg. 14-16, Table 3: might a good example "process" for the breakdown illustrated here be the biological pump itself?; by looking at this, the authors could (a) give an example of how conventional understanding of a carbon cycle process can be translated into their conceptual model, and (b) illustrate the difficulties of doing so because of the associated complexity, and (c) thus emphasise the importance of a holistic viewpoint

**We did not incorporate the example of the biological carbon pump in table 3 but centred our discussion around that example.**

Pg. 17, ln. 298: most existing models already incorporate this cyclic aspect; what's special here?

**We deleted that concrete sentence and rewrote the discussion.**

Pg. 17, ln. 299-301: this sentence may describe how the process of export is *sometimes* described, but I think it's an exaggeration to suggest that this is "normal"; most modelling scientists are well aware of return pathways; to be honest, return pathways can even be highly visible at the surface, e.g. CO2 outgassing along the equatorial Pacific

**We rewrote the discussion.**

Pg. 18, ln. 342-345: where this conceptual model seems useful to me is in pointing to fluxes, and making it clear where they sit in a wider consideration of the cyclic pathways of carbon

**We added this feedback to our motivation and new reasoning chain.**

Pg. 17, ln. 301-303: being (as the authors keep stressing) purely qualitative, the conceptual model under discussion here cannot really help with this point much more than making it clear (which may, as already noted, be obvious for many researchers) that "what goes down must come up"

**We rewrote the discussion and do not stress the qualitative character of our concept anymore.**

**Changes made to comments of referee 1:**

**For the following comments, please consider that coming from a numerical model community, the word 'model' is generally associated with equations and numerical output in my mind. While the term 'model' can be used for many purposes to define a representation of reality, I suggest that you should present a 'visual-model' in opposition to 'numerical-model', to avoid to any other biased modeler as me to look for equations and numerical outputs **

**We changed the wording to "concept" throughout the manuscript.**

Title: While publishing in 'Ocean science', the term 'marine' can appear in the title. Following the first referee's comment, I suggest putting the table directly with the information about your work 'a visual-model for mapping...' .

**We changed the title to: There and back again, a journey of many pathways: conceptualising the marine organic carbon cycle**

Abstract:

p1.Ln 3 : 'other related tasks' sound vague, so I suggest removing it.

p1.Ln 3 : you may use 'cycle' instead of 'pathways', as in the line right after you are adding 'processes and pathways' ?

p1.Ln 3 : 'qualitative-visual' model can be mentioned instead of qualitative model only.

**We rewrote the entire abstract.**

The first paragraph (p1 Ln15 to 21) is difficult to follow without the table placed. I suggest starting with your current second paragraph (p2.Ln22 to 29) and eventually move/rewrite this first paragraph with the last ones of the introduction where you talk about your work.

**We rewrote the introduction and started with a more general paragraph on organic carbon pathways (p 1 first paragraph).**

p2.Ln22: Instead of 'marine ecosystems and the OC cycle', can it be reduced to 'marine OC cycle' directly ?

**We shifted that paragraph into the discussion and reduced it to 'OC cycle' (p20 L418).**

p2.Ln22 to 24 : Why do you focus only on particles on the surface ? You can generalize as 'An OC particle in the ocean can end up …'

**We keep the spatial restriction here (p20 L418), as we have specified that all the pathways we consider start in the surface waters.**

p2.Ln23 to 24 : I suggest to re-write to have only one sentence, e.g. ' Each pathways is unique in its sequence of processes, and there is a myriad of them' . As there are a myriad of pathways AND processes.

**We rewrote and shifted this part: Each OC compound travels its pathway through the OC cycle. An OC compound in the surface ocean may end up on the surface or in the deep sea, be decomposed, or become recalcitrant, to name just a few possibilities. Each pathway is unique in its sequence of processes. So, there is a multitude of possible pathways.  (p20 L417-420).**

p2.Ln36 : I suggest to add 'vertical export flux' instead of only 'export flux'

**We deleted this part of the introduction.**

In these two paragraphs (p2.Ln22 to Ln 37) some link to the feedbacks/conclusions made through those existing tools with the climate can be made, to reinforce the use of this visual-model for further use.

**We shifted some of the paragraph (p2 L22-29 to p20 L417-420) and deleted paragraph p2 L29-37.**

Ln39-40: I suggest to temper what it is mentioned by providing generalities such as ' ..this destination is mainly considered by changing the particles' **properties (e.g. density, shape)**'. In this case some additional references may be required.

**We deleted this section.**

p2.Ln40-42: I would be glad if you have in hand a reference to add, that points out these facts.

**We have added examples of the described small inconsistencies in visual representations of the pathways and inserted the whole paragraph into the new chain of reasoning (p.2 L47-57).**

p3.Ln82 : I do not see the point to add the notion of coral reef here.

**We deleted it.**

p3.Ln84 : I would have been glad to have already the information that table 1 will provide me with the dictionary of the nomenclature used in the following part. I suggest adding a sentence letting the reader know about Table 1 before moving to the explanation, as the sentence in p2. Ln84 does not sound clear to me.

**We have added an insertion: Given that we conceptualise only the OC pathways (for a definition of relevant terms of the concept, see Table 1), we do not resolve carbonate and alkalinity interactions, and do not display marine carbonate systems within our concept (p2 L84).**

p3.Ln85 : ' (POC) **embedding** living and non-living OC **particles**'

**We changed the sentence as suggested (see p4 L103).**

p3.Ln85 : Thank you for the correction made following Referee's 1 comment regarding the size mentioned/used.

**No changes made.**

p4.Ln.121: 'under the given hygiene conditions'; as we are already in an analogy, barely used in scientific writing, I suggest to restrain the other reference to current society behavior, as in a number of years when people will refer to this manuscript they may not be able to understand the reference to the current pandemic situation as easily as us today. I suggest deleting the allusion.

**We deleted the sentence.**

p5.Ln127-151 : Be sure that the words used here are consistent with the eventual reswamp of the nomenclature (see my comments below about table 1). This paragraph is really hard to follow even if I greatly appreciate the effort to place a nice analogy for explanation.

**We have rewritten the section (p. 6-7). We have also adapted Table 1 (p. 5) to make it more accessible and added a methodological diagram as a supplement.**

p7.Ln154 : The repetition of 'spatial, spaces, spatially' can be avoided (e.g. ' By defining four spatially bounded volumes with…').

**We changed the wording: Hence, pathway patterns cannot be unambiguously defined without spatial information. To systematically add this information, we define five spaces, volumes with distinctly different environmental conditions and processes (p.7 L172).**

p7.Ln155 : the reference to Table 1 is not informative and necessary.

**We deleted it.**

p7.Ln163-164 : I am curious why these specific systems must be represented with these specific numbers of spaces ? Is it to be sure to consider the processes/pathways in these specific systems ? As there are no 'numerical' rules for the conceptual model here, I am wondering why this information is here.

**We have added that this rule applies in the case that all closed loops are resolved (p.7 L182).**

p7.Ln154-164 : I am wondering why the Atmosphere is not considered here as the LSS is ?

**We added an atmosphere space (AS).**

p10.Ln209 : It is confusing that 'pathway patterns' means closed loops. Why not use closed loops directly ?

**We have rewritten the first two sentences of this paragraph (p.10 L235).**

p17.Ln311 : Not only fish and mammals, but also reptiles (e.g. McClain CR, Nunnally C, Dixon R, Rouse GW, Benfield M (2019) Alligators in the abyss: The first experimental reptilian food fall in the deep ocean. PLoS ONE 14(12): e0225345. https://doi.org/10.1371/journal.pone.0225345)

**We have added the study to the relevant paragraph (p.19 L399).**

p18.Ln334 : 'biological**carbon**pump'

**We changed it throughout the paper.**

For Initial position I do not get the meaning of 'Abstract' in the definition. Would 'Start position' will suffice ?

**We keep "abstract" because we also mention on page 3 L83 that loops have no initial point, and that the choice of such a starting point is thus abstract.**

In the example part, I suggest putting in bold the terms to have an easier reading but to not use the example of the term defined above. It is confusing to have in the example of one term and example of the term above. One should be able to see the example only jumping from one line to another in the example column.

**We have adjusted the definitions and added three concrete pathways at the beginning to additionally show how we got from concrete process-based pathways to our abstract definitions of pathway patterns (p.5). We also added a flowchart of the methodology as a supplement B.**

Process: In the example column you can add 'fish respiration'

**We have adapted Table 1, taking into account the reviewers' criticism of the example column and the problems in distinguishing the terms.**

Path segment: In the example column, you can delete the 'Processes line example', remove the 'path segment' and keep only 'OC remineralization (..)'.

**See comments above.**

Pathway: In the example column, I do not get why OC remineralization (presented above as Path segments) is now considered as a 'sequence of path segments'. Are the pathways defined as 1) how the carbon moves from one 'box' to another along processes, or 2) how the carbon moves in the conceptual space volumes?

**We adapted table 1.**

Space: This is not necessary for me to be defined here.

**We keep this definition here, as the table should provide all relevant information.**

Closed loop/Open loop: To the current state I unfortunately do not get clearly the distinction between pathway or loop. There is no need for two pathway examples here, it leads to confusion between pathway and loop. You can stick only with 'surface remineralization loop'.

**We adapted table 1.**

Process option: While I get that you want to define all the process options that we know, I suggest that the options should be already included in the 'conceptual' processes. By itself each process option is a process.

**We agree and changed it to processes throughout the manuscript.**

Pathway pattern: Similarly I get confused with the distinction.

**We adapted table 1, added a methodological concept figure as supplement B and revised section 3.1.**

Other proposition: To help the reading between the text of the manuscript and the table, you can refer as example to the same example available in the text such as for Path segment (mention the six critical path segment of the OC cycle (p5.Ln37-39) ) ; for Open loop ( mention the five ones (p7.Ln165)) ; and for Close loop ( mention the 3 ones (p9.Ln202-206)).

**We adapted table 1 with three example pathways that are part of the defined closed and open loops.**

To help to connect with the text, Is it possible to have a specific code in the legend and the figure for the loops and one for the path segments?

**Following referee 1, we added a flow syntax to the publication where helpful and necessary. In the figures, path segments are indicated by arrows, critical path segments by capital letters and loops by a colour code. We added this information where it was missing.**

The atmosphere term should be appearing as the long-term sediment one is appearing.

**True, we added it (first introduced p.7 L181)**

For the nomenclature see one of the last comment made below for the link between Table 3 and Fig2.

**We eliminated inconsistencies between section 2, table 3 and figure 2.**

I suggest deleting the repetition of the column name each time we are moving to another path segment and therefore place the column name as the first line before the firth path Organic Cabron position change (A).

**The suggestions did not help the flow, so we kept the original structure.**

I suggest to either place in the center the name of the path to cut the reading among the table each time the path changes, and/or use a double line before and after the name of the path, similarly to help the eyes to see that we are moving to another path.

**We added spaces and double-lines to allow a more structured reading of table 3.**

On the part of the table in p16, I do not see why we have again the path C, and why the path E is reduced only as a name in the first column? I guess it is a typo and the following processes refer to path E following the infos available in Figure 2.

**We modified table 3 eliminating inconsistencies and adding missing path segments.**

I suggest removing the column Process description as it is already well explained in the text and in figure 2, as well as the column Involved organisms. It will save space and help the reading. If you want to keep one among the two I suggest keeping the process description one.

**No changes made.**

However, an effort can be made to smooth again the nomenclature used between Figure 2 and Table 3 First column, to allow the reader to proceed to an easy retrieval of the processes (description or representation) between the Figure 2 and the Table 3.

**We eliminated inconsistencies between section 2, table 3 and figure 2.**

Organic Carbon Position Change: Following Fig.2 it seems that the biotic direct transport is not writing in the process column while infos related to seem appearing in the other columns (?)

**We added direct biota-induced transport (p.13).**

OC Remineralization (D): Following Fig.2 it seems that the DOC consumer respiration is not writing in the process column

**We added DOC consumer respiration (p. 14).**

Editorial/Typo comments :

p2.Ln49 : 'DOC' acronym has not been defined yet.

**The DOC acronym is now defined before the acronym is used (p.4 L104). Mentions of dissolved OC before are written out.**

p2.Ln52 : As defined in the p2. Ln49, you can use the acronym

**Part is now on page 2 line 32 and not defined before.**

p3.Ln60 : Extra parenthesis

**p3.Ln60 : Part is deleted.**

p3.Ln65 : Missing parenthesis

**p3.Ln65 : Part is deleted.**

p5.Ln126 : Missing a dot at the end of the sentence

**Full stop is added (p.6 L132)**

p7.Ln167-176 + other parts in the text : To help the reading of acronyms, can you think about having the related space ones (SLS,WCS,USS) in italic and the one related to the loop in normal?

**We highlighted acronyms in italic when we mention them for the first time.**

p10 to 13 : 1) Can we have subtitles for each path segment you are talking about ? Like it is done clearly in Fig.2 (Path segment A, Path segment B, etc. ) ? 2) Can we have just one sentence at the beginning referring to Figure 2 and Table 3 for this entire part, instead of having it mentioned everywhere ? It will make the text easier to read.

**We do not add subtitles to not disturb the text flow but table 3 is the structured summary of the text.**

**We referred to table 3 and figure 3 in the beginning of the section and reduced the references in the following text.**

p11.Ln253 : Extra parenthesis

**Extra parenthesis is deleted (p.15 L283).**

p18.Ln336 : 'e.g.' before via

**We deleted this part.**

References:

Arrieta, Jesús M.; Mayol, Eva; Hansman, Roberta L.; Herndl, Gerhard J.; Dittmar, Thorsten; Duarte, Carlos M. (2015): Ocean chemistry. Dilution limits dissolved organic carbon utilization in the deep ocean. In Science (New York, N.Y.) 348 (6232), pp. 331–333. DOI: 10.1126/science.1258955.

Dittmar, Thorsten (2015): Reasons behind the long-term stability of dissolved organic matter. In Dennis Hansell, Craig Carlson (Eds.): Biogeochemistry of marine dissolved organic matter: Elsevier, pp. 369–388.

Hansell, Dennis; Carlson, Craig; Repeta, Daniel; Schlitzer, Reiner (2009): Dissolved Organic Matter in the Ocean: A Controversy Stimulates New Insights. In oceanog 22 (4), pp. 202–211. DOI: 10.5670/oceanog.2009.109.

Hansell, Dennis A. (2013): Recalcitrant dissolved organic carbon fractions. In Annual review of marine science 5, pp. 421–445. DOI: 10.1146/annurev-marine-120710-100757.

Jannasch, Holger W. (1967): GROWTH OF MARINE BACTERIA AT LIMITING CONCENTRATIONS OF ORGANIC CARBON IN SEAWATER. In Limnol Oceanogr 12 (2), pp. 264–271. DOI: 10.4319/lo.1967.12.2.0264.

Jiao, Nianzhi; Herndl, Gerhard J.; Hansell, Dennis A.; Benner, Ronald; Kattner, Gerhard; Wilhelm, Steven W. et al. (2010): Microbial production of recalcitrant dissolved organic matter: long-term carbon storage in the global ocean. In Nature reviews. Microbiology 8 (8), pp. 593–599. DOI: 10.1038/nrmicro2386.

Kharbush, Jenan J.; Close, Hilary G.; van Mooy, Benjamin A. S.; Arnosti, Carol; Smittenberg, Rienk H.; Le Moigne, Frédéric A. C. et al. (2020): Particulate Organic Carbon Deconstructed: Molecular and Chemical Composition of Particulate Organic Carbon in the Ocean. In Front. Mar. Sci. 7, p. 518. DOI: 10.3389/fmars.2020.00518.

Mentges, A.; Feenders, C.; Deutsch, C.; Blasius, B.; Dittmar, T. (2019): Long-term stability of marine dissolved organic carbon emerges from a neutral network of compounds and microbes. In Sci Rep 9 (1), p. 17780. DOI: 10.1038/s41598-019-54290-z.

---

## Referee Report (RR1)

Round 2 - Review prepared by Gwenaëlle Gremion

PostDoctoral fellow at the Université du Québec à Rimouski, Canada
* * *
for Scheffold & Hense, "**There and back again, a journey of many pathways: conceptualising the marine organic carbon cycle**"
* * *
**Summary:** Please Refer to **https://doi.org/10.5194/os-2021-75-RC2**

**Decision:**

I acknowledge the work done since the first version of the manuscript, which greatly improves the value of the work presented. Some substantial suggestions follow, but I am in support of the manuscript moving forward for publication after consideration of some of the comments.

**Substantial comments:**

*Abstract* **:**

p1.Ln 3 : Suggestion : 'with sophisticated **tools and ** mainly **by** quantitative methods [..]. '

p1. Ln 6 : What are the significance of 'core structure' and 'sub-concept' ?

p1. Ln 8 -10 : I suggest combining the two sentences such as e.g. ' In response, we propose a (visual) concept that defines pathway patterns who are defined by mapping, comparing…based on a consequent literature review' .

p1. Ln 10: To be consistent it should be a closed-loop 'pathway' as 'patterns' is used later for defining rDOC and remineralization. But as it is the abstract I suggest sticking to open and close loops only.

p1. Ln12 : As it is really technical in terms of vocabulary, every word used has a meaning, I suggest writing 'loops' instead of 'basic structures'.

p1. Ln13: I suggest adding 'carbon' before 'pools' and I am not sure of the meaning behind 'agent' at this stage.

p1. Ln 15 : Do we want to stay large and talk to OC cycle, or do we want to specify 'marine OC cycle' in this explanation ?

p1. Ln 16: As before:  What is the significance/definition of 'core structure' ?

p1. ln 17 : Are we sure 'basic' is needed here (and in the following sentence). I suggest reducing the wording as much as possible to avoid confusion.

*Introduction* **:**

It is a nice improvement of the first version of the manuscript.

p1.Ln 22 : Suggestion : '..OC dynamics **along them ** is an essential and **relevant** focus on ocean research.' Instead of 'OC dynamics resulting from the multiplicity of these pathways and the human influence on them is an essential and very productive focus of ocean research'. As I am not sure the human influence is the main topic of this paper, and as I do not see how a focus can be productive.

p1. Ln 24-27 : I am not sure I understand the sentence. Is it the comprehensive observations and the sophisticated numerical models who improved the carbon budgets ? Maybe consider a rephrasing of the sentence.

p2. Ln 29 : At this stage the definition of higher-level structures, core mechanisms is not intuitive. I suggest sticking with what will be used after ( Pathways and sequence of processes).

p2. Ln 29 - 31 : Suggestion : ' ..of the OC cycle,  the studies focus only on the description of pathways related to the interest of the research". Instead of '... OC cycle, these concepts have a relatively narrow focus and consider a selection of pathways.'

p2 . Ln 31-34 : Similarly as comment for p1. Ln 24-27 , the utilization and referencing of the example make it hard to understand. Is it an enumeration, or one sentence only ? Maybe consider a rephrasing of the sentence.

p2. Ln 43 : Suggestion : ' useful' instead of 'plausible' ?

p2. Ln 48 : Why do we have 'graphics' twice in the sentence ?

p2. Ln 50 - 53 : Suggestion : ' For example, Steinberg and Landry (2017), Cavan et al. (2019), Anderson and Ducklow (2001) and Boscolo-Galazzo et al. (2018), while aiming to represent the same pathways do not use the same visual representation leading to inconsistencies.  As  the aim of such studies is not to create congruent conceptual representations of the OC cycle, their visualizations are still useful tools to highlight their research focus in an overarching picture. '

p2. Ln 57 : Suggestion : ' Non-congruent graphics within the scientific literature to represent a same concept do not exploit the full potential'

p2. Ln 58 : Do we want to stay large and talk to OC cycle, or do we want to specify 'marine OC cycle' in this explanation ?

p3. Ln 65 : Do we want to stay large and talk to OC cycle, or do we want to specify 'marine OC cycle' in this explanation ?

p3. Ln 68 : I suggest removing core to avoid confusion on the definition associated with 'core similarity' that may not be clear at that stage of the manuscript.

p3. Ln 73 : Do we want to stay large and talk to OC cycle, or do we want to specify 'marine OC cycle' in this explanation ?

p3. Ln 71-75 : I really appreciate this paragraph. It is well structured and gets straight to the objectives of this study. It is a nice addition to the first version of the manuscript.

p3. Ln77and 78, 79 : I suggest removing core to avoid confusion on the definition associate with 'core similarity' that may not be clear at that stage of the manuscript.

*Part 2 :*

*I greatly appreciate the modifications made for this part regarding the first version of the manuscript.*

Table 1 :

*Space :* To highlight that you are considering 'Atmosphère', 'Ocean' and 'Sediment' I suggest to list all your 5 spaces in the example column. Suggestion : Atmosphere Space (AS), Ocean spaces (e.g. Surface layer space (SLS) and Water column space (WCS)) ; Sediment spaces (e.g. Upper (USS) and Lower (LSS) sediment spaces). '

*Initial position :* Suggestion for the Example column, to use the same wording : .. in the ** Surface Layer Space**  instead of surface space.

For me the terms 'pool' , even intuitive, should be described as well as agent (not as intuitive) that you use several times in the abstract and Introduction.

My understanding is that 'pathway patterns' and 'sub-patterns' are the same thing. I suggest removing the 'sub-pattern' wording here and in the following text to avoid any confusion in the wording.

I do not think the first line of the table with the Mapped example pathways in the base pathway concept is informative, it leads more to confusion in my point of view.

*Part 3 :*

p4. Ln 114-115 : This sentence is not useful or can be merged with the first one.

p4. Ln 115-116 : Maybe you can try to have a logical order when listing the spaces : up to down (Atmosphere - surface - sediment) or down to up (sediment - surface - atmosphere) .

p4. Ln 117 - 118 : This sentence is really hard to understand as a lot of things are mentioned with no clear definition or point of difference : 'base pathway', 'mapped pathways', 'core structures', 'core patterns of OC pathways'. It is really hard to get the nuance among the notions. Maybe the 'base' pathway concept can be named as 'litterature-based-pathway-concept', 'mapped pathways' which are the ones you are describing can be named simply 'OC pathways', and I do not get the sense and distinction  of core structures/patterns.

p6. Ln 123 : Following my previous comment, you can use the appropriate appellation ' To explain how to compare and condensed litterature-based-pathway-concept and define'

p6. Ln 123 : Once more, what is the 'core patterns' meaning ?

p6. Ln 140 : I do not get the meaning of the 'entire-city-beach route'. Following your explanation it should be named ' harbor front beach route'  otherwise I do not get why the harbor front beach route is a subordinate of the entire-city-beach route' as it is the same thing.. ?

p6 Ln 144 : Please be consistent in the wording. What route is referring to here ? Path segments or Pathway patterns ?

p6. Ln123-147 : From the explanation you provide, I drew a schematic (Schematic 1). But it seems that the term 'pathways' is not properly placed in my schematic. Maybe it is my understanding wrong, or maybe something is misleading in the explanation. I'll let you have a second look on the text to be sure.

[Figure]

*Schematic 1 : Understanding of the analogy with the terminology mentioned in the Text.*

p8. Ln188 : For consistency with my comment about Table 1. 'sub-patterns' should be replaced by 'pathways'.

p8. Ln 221 : At the end of the sentence, Are we sure the wording is sub-pathway patterns and not 'pathway patterns' ?

p9. Ln 229 : for consistency it should be 'pathways' and not 'sub-patterns' (See comment on Table 1).

p7. Ln 160 : As you already said in Sect 2 this, I recommend using wording such as ' As previously mentioned, the path segments…' .

p10 Ln235 to p16 Ln 330 : For consistency with the italic used to characterize pathway and path segments, can we place the processes in italic in the text too ?

p12 Ln. 253 : 'A of (r) DOC in 2)' . Does the '2' refer to Fig.2 ?

p15 Ln. 287 : I suggest the reading of this paper to add a reference here :Goldthwait, S., Yen, J., Brown, J., and Alldredge, A.: Quantification of marine snow fragmentation by swimming euphausiids, Limnol. Oceanogr., 49, 940–952, https://doi.org/10.4319/lo.2004.49.4.0940, 2004

p15 Ln. 294 : The wording here may be misleading. 'sub-patterns' should be pathways, and POC-DOC remineralisation 'sub-loops' ?

p16 Ln. 325 : Instead of sub-pattern, shouldn't it be 'sub-loop' ? (See my schematic 2).

[Figure]

*Schematic 2 : Schematic made following explanations and my understanding of the text.*

p16 Ln.327 and 329 : Does the term 'sub-patterns' refer here to the pathways or to the sub-loops previously mentioned and called-sub-patterns before ?

p16 Ln. 330 : Does the term 'patterns' refer here to pathway patterns, or the previous term sub-patterns that are confusing (see previous comment) ?

Figure 1 :

- For consistency it should be 'pathways' and not 'sub-patterns' (See comment on Table 1) in the legend.

Table 2 :

Uniformize the term 'sub-pattern/Pathways' (See previous comments on that point).

Table 3 : *It is a great improvement of the first table 3 proposed in the first version,  congratulations !*

-May I suggest to remove the repetition of the column names (Process, loop syntax, etc.) for each Path segment. The reader may refer to the first one if s/he needs a reminder. Therefore I suggest to place the column names above the first path segments to be clear these names apply for the entire long-table.

-I am wondering if we do not have the same information twice with Fig 2 and Table 3 ? Do we want to keep both, or do we want to choose one of them ? Just a thought.

*Discussion :*

Entire discussion : To avoid the repetition of the main reference Giering and Humphrey 2020, maybe in the second paragraph of the discussion you can make a statement that mentions that in the following analysis the description of BCP used as reference is based on Giering and Humphrey 2020 ? With this statement the reader will know that further assumption will refer to their work and you would not have to mention it in every paragraph ?

p17 Ln 342 : This sentence is a repetition of the previous paragraph. It should be reswamp if you want to keep the information that relates with your example (F [SLS]).

p17 Ln 351 and 352, 353, 357, 358 , 360: As mentioned before, sub-patterns should be removed and sub-loop should be used (See my schematic 2).

p17 Ln 354 and 358 : Does 'loop' refer to the sub-loop ? If yes please use consistent wording.

p17 Ln346 to 363 : It is not really clear when reading the text and having the Figure 3 under the eyes how the number of loops is determined. In p17 Ln. 348 It is mentioned 'to close the loop' inducing that there is one loop in the panel (a) is misleading of what it is stated at p17 Ln 354 when ' only two loops of the superodinate loops of panels (a)'. Is it possible to have the number of loops associated with the pathway patterns mentioned in the Figure 3 legend ? Similarly, the loops are not easy to see on the Figure pannels, and when the author refers in the text to seven loops or six loops form panels (c) and (d) it is misleading with the numbering of pathway patterns mentioned in the Figure. I suggest either talking only of pathway patterns numbers in the text to fit with the legend of the Figure, or to switch the legend of the figure with the numbering of loops to fit with the text.

Figure 3 :

- The space (SLS) and (WCS) can be placed once on the left side of the figure.
- The term 'Processes' can be placed in bold above biota-induced and physical
- The term 'loops' can be placed in bold above the various loops specified

*Supplementary material A :*

I really acknowledge the work placed in the construction of this huge diagram, I know the work it asked for and how difficult it is to synthesize it. However, it may be scary at first look. It may look like the electrical scheme of the space shuttle for someone who would like to read/use it, and I would like to provide some ideas to maybe improve the visibility of this work, as I was not able to review it properly due to its complexity and difficult visibility. However, as it is a supplementary material I do not see it as a mandatory requirement for publication of this work if you want to keep it that way.

- First Review the arrow legends, as some are placed below the arrows and are sometime difficult to read (e.g. Coastal in the sediment part 'Consumed macrophytes' below the black arrow) ;

- Some of the text are missing space between words (e.g. Coastal 'Carnivoresand detritivores')
- Some of the arrow descriptions are similar, maybe you can manage to have the same infos placed where the arrows merge ? (e.g. Coastal, Sinking of resting stages).
- Why don't' you use boxes for Fecal pellets,bacteria and Virus ?
- As you refer to 'benthic carnivores', what imply 'Benthos' ?
- Why only referring to mammals ? You may refer to the upper trophic levels to be as general as possible ?
- Shouldn't 'Pahotrophy' be Phagotrophy ?
- Physical Transport is written numerous times (with some typo (Phyisical)), but it is already linked to blue arrows that are mentioned in the legend as physical-induced, so maybe there is no need to write it down ?
- Does 'Autigenic' shouldn't be Authigenic ?
- Is it possible to have this huge diagramm 'interactive' ? Is it possible to have in the legend the SLRL loop display for example, and when someone click on it is only the SLRL arrows and boxes and infos that appear for a better visualization?

*Supplementary material B:*

- In the box 6, as Pathway pattern abbreviation has already been described in box 4 you can either use the full wording or the abbreviation only but not both, it is confusing.

**Editorial/Typo comments :**

p8 Ln 189 : Do not use italic for and between the two sub-patterns/pathways.

p10. Ln 232 : Do not use italic for and between the two sub-patterns/pathways.

Legend Figure 1 : "loop" when talking about srDOCL and LrDOCL shouldn't be plural ?

Legend Table 2 : "loop" when talking about srDOCL and LrDOCL shouldn't be plural ?

p12 Ln. 249 : Even if it is the beginning of the sentence, I suggest to force the r of (R)DOC to be in lowercase.

p12 Ln 253 : The path segment A should be placed in parenthesis.

p12 Ln 258 : The path segments A and E should be placed in parenthesis.

p12 Ln259 + all the manuscript+Figures/Tables : Shouldn't be '(r) DOC' instead of rDOC ? Maybe I am confusing the meaning, but please review all the manuscripts and supplementary material if the wording with and without parenthesis means the same thing. If not please mention somewhere the difference between the two ways of writing it.

p12 Ln 267 : The path segment A should be placed in parenthesis.

p12 Ln 272-273 : The path segments A and E should be placed in parenthesis.

p12 Ln 277 : The path segment D should be placed in parenthesis.

p15 Ln 281 : The path segment D should be placed in parenthesis.

p16 Ln 303 : The path segment D should be placed in parenthesis.

p17 Ln 348 : The path segment E should be placed in parenthesis.

p17 Ln 351 and 354 : The path segment E should be placed in parenthesis.

Figure 3 :

- I wonder if the figure 3 would be better if seen as landscape instead of portrait within the page ?

p19 Ln385 : The path segment E should be placed in parenthesis.

----- Thank you -----

---

## Author Response (AR2)

First of all, we would like to thank referees 1 and 2 again for their time and constructive criticism. Below we have listed the comments and our responses in tabular form.

**General comment on terminology (C1):**

Apart from the changes we have highlighted in the table, we decided to adjust our terminology as the referees' comments made clear that it was still too clunky and led to misunderstandings, probably partly because of the terms used.

In order to reduce the complexity of our concept and improve the terminology, we have changed the following points:

- Deletion of sub-pattern and critical - these terms are not necessary. We have thus reduced the terminology
- Renamed **path segment to functional segment**
- Renamed **pathway pattern to structure**
- Renamed base **pathway concept to literature-based pathway concept**

In addition, we have streamlined the explanations in the text and reduced unnecessary additions to reduce the complexity of the manuscript.

**Answers to referee 1:**

| Comment referee 1 | Answer | New line in manuscript |
|---|---|---|
| Ln. 2: "of our time" -> "in oceanography"; this is secondary compared to other aspects of the Earth system's carbon cycle | We changed it to " one of the pressing tasks of our time". | 2 |
| Ln. 3: "investigated" - by who?; it should be clear whether or not this manuscript is meant here; for instance, you could write "... are typically investigated ..." since this makes it clear you're talking about the general situation rather than your specific one | We agree and change it to "are typically investigated". | 3 |
| Ln. 10: "scales" -> "distributions"? | We changed it to: In response, we propose a (visual) concept in which we define such higher-level 'structures' by comparing and condensing marine OC pathways based on their sequences of processes and the layers of the marine system in which they operate. | 8-10 |

| | | |
|---|---|---|
| Table 1: "sub-pattern" - a somewhat confusing term | We agree and deleted the term in Table 1 and changed the sentences following also C1 to: A structure that comprises all pathways returning to the initial position is named closed loops. A structure that comprises all pathways not returning to the initial position is named 'open' loops. | Table 1 |
| Ln. 235: but what do you mean by "superordinate" itself? | We changed the sentence to: Having defined the structures of remineralisation and rDOC loops, we now… | 232 |
| Ln. 282: "allochthonous" - I find that "external" and "internal" origin are maybe clearer than these jargon terms | We keep the terms, as allochthonous is not a jargon term but a technical term that describes rather the source of origin. External or internal might be misleading. | |
| Ln. 286: "indirectly" - Or is it directly? Mayor et al. suggest this may be a strategy ... doi: 10.1002/bies.201400100 | We thank the reviewer for his comment and the interesting paper. We refrain from judging whether the processes are indirect or direct and changed the sentence to: Consumers reduce the size of organic POC by sloppy feeding on living and non-living POC by e.g. zooplankton coprorhexy (Lampitt et al., 1990), by producing small metabolites, by excreting DOC (Lampert, 1978) or by swimming or moving (Dilling and Alldredge, 2000). | 279-282 |
| Ln. 295-298: the double brackets in this paragraph are correct, but they're distracting! | We agree and changed the sentence to: In addition, bacteria can oxidise VOCs and $CH_4$ as e.g. shown in Halsey et al. (2017) (D of VOCs/$CH_4$ in Figure 1). The VOCs and $CH_4$ origin from abiotic processes such as photochemical degradation of DOC (Kieber et al., 1989) and biogenic processes, e.g.production by phytoplankton (Lenhart et al., 2016) and zooplankton in anaerobic areas of their guts (Weber et al., 2019; Schmale et al., 2018). | 289-292 |

| | | |
|---|---|---|
| Ln. 317: "recalcitrant" - this could be clearer; it's only seemingly "recalcitrant" because its concentration is too low for the relevant degrading organism to make a good living breaking it down; this is quite different from genuinely recalcitrant material that is bioengineered to be difficult to destroy (e.g. lignin, sporopollenin); spell this out to help your readers | We changed the sentence to: Furthermore, processes that convert living and non-living POC into DOC, e.g. dissolution, can dilute DOC to the point where it can no longer serve as sufficient nutrition for microbes and can be considered technically recalcitrant (Arrieta et al., 2015) (Figure 2, arrow from POC to rDOC). | 313-315 |
| Ln. 324: "of higher resolution" -> "with greater complexity"; "resolution" may carry some spatial context | We retain the term "resolution" because we do not think it has a general spatial connotation, since, for example, the resolution of photographs means that more pixels and thus more information are shown without any spatial context being associated with it. | |
| Ln. 329: "discussion" - would it make more sense to put the example in its own "results" (or "example") section or something?; and then use the discussion section to more distinctly discuss the framework | Although we understand the referee's point of view, we will not separate example and discussion because we have interwoven the two in the discussion and do not think we can reasonably separate one from the other. Especially since our discussion is mainly based on the example. | |
| Ln. 332-333: "embedded processes, pools, and agents" - these terms need to be clearly defined from the outset; "agents" is only formally defined in section 3.1, and then passingly in a bracketed clause; having something upfront about what is meant in each case would be helpful for some readers | Process is defined in Table 1. We added a definition for agents (organisms that initiate or execute a process) and pools (reservoirs of a certain substance- in this case organic carbon. Pools can be non-living and living). | Table 1 |
| Ln. 334: "consistent terminology" - a consistent terminology would be good; the one here satisfies this, but its clunkiness may doom it | We agree with the referee that a uniform terminology is necessary, and we also see that our proposal is certainly only a first step towards finding this terminology but hopefully stimulates a discussion. However, we adapted our terminology to decrease complexity see C1. | |
| Ln. 337-341: brilliant!; this really helps - thanks! | We thank the referee for the comment. | |

| | | |
|---|---|---|
| Figure 3: an argument could be made for showing what the solubility pump looks like in your diagrams and/or terminology; admittedly (a) it's inorganic carbon, and (b) it'd be super-boring for sure compared to the biological pump, but in making this clear your system could demonstrate some value I think | Although we appreciate the ideas of referee 1 as they were very helpful for including the example of the biological pump in the first place, we will not include an additional example for the solubility pump. Mainly because of referee 1's argument that the manuscript is already complex and demanding. Adding another level would make the manuscript even longer and would - in our view - not add enough value to justify extending the manuscript. | |
| Ln. 345-346: "Missing further information, …" - not sure what "missing further information" means here; expand or delete | We changed it to: As it is not clarified in the definition. | 343 |
| Figure 3: It would be difficult to make tidy, but I might be tempted to put titles on each of these panels; e.g. (a) is the "summary" or "overview", (b) is "resolved pools", (c) is "biological transport" and (d) is "physical transport" | We added: a) BCP as defined, b) Resolved pools c) Direct biota-induced transport and c) Physical processes. | Figure 3 |
| Ln. 373: "sensu stricto" - more confusing Latin; expand for clarity; also, while there is something of a point about gravitational sinking not in itself being biological, and therefore arguably separate from the biological pump, I might be inclined to skip this here as it only adds confusion to an already difficult to follow manuscript. | Although we think that the question of whether sinking by gravity is part of the biological pump is an interesting point, we agree with the referee that this debate is not relevant to our discussion. We deleted the sentences. | |
| Ln. 397: "neglected" - is it "neglected" or is it simply viewed as "secondary" on a quantitative basis?; where a process is not considered important, it is often "neglected" in experiments or models for simplification | We agree and changed the sentence to: was considered quantitatively secondary and therefore neglected. | 387 |
| Ln. 609: "Theoceans" -> "The oceans"? | We adapted the reference. | 602 |

**Answers to referee 2:**

| Comment referee 2 | Answer | New line in manuscript |
|---|---|---|
| p1.Ln 3 : Suggestion : 'with sophisticated **tools and ** mainly **by** quantitative methods [..]. ' | We refrain from changing it, as this would not add value to the sentence, but prolong it. | |
| p1. Ln 6 : What are the significance of 'core structure' and 'sub-concept' ? | We changed it following C1 to: Such structures can provide a framework for the growing number of partly overlapping concepts, which conceptualise selected OC pathways, and promote more structured comparisons and consistent communication, especially between different disciplines. | 6-8 |
| p1. Ln 8 -10 : I suggest combining the two sentences such as e.g. ' In response, we propose a (visual) concept that defines pathway patterns who are defined by mapping, comparing…based on a consequent literature review' . | We changed it following C1 to: In response, we propose a (visual) concept that defines such higher-level 'structures' … | 8-10 |
| p1. Ln 10: To be consistent it should be a closed-loop 'pathway' as 'patterns' is used later for defining rDOC and remineralization. But as it is the abstract I suggest sticking to open and close loops only. | We changed it following C1 to: The resulting structures comprise 'closed loops', three remineralisation and two recalcitrant dissolved organic carbon loops that close in marine systems, and 'open loops',… | 10-12 |
| p1. Ln12 : As it is really technical in terms of vocabulary, every word used has a meaning, I suggest writing 'loops' instead of 'basic structures'. | We agree and changed it to: In addition, we provide a synthesis of embedded processes, OC pools, and process-executing organisms (agents) embedded in these loops. | 12-13 |
| p1. Ln13: I suggest adding 'carbon' before 'pools' and I am not sure of the meaning behind 'agent' at this stage. | Pools and agents are now additionally defined in Table 1. We also changed it to: In addition, we provide a synthesis of embedded processes, OC pools, and process-executing organisms (agents) embedded in these loops. | Table 1 |
| p1. Ln 15 : Do we want to stay large and talk to OC cycle, or do we want to specify 'marine OC cycle' in this explanation ? | We added marine in all cases where we do not refer to the overall carbon cycle. | 16 |
| p1. Ln 16: As before: What is the significance/definition of 'core structure' ? | We deleted 'core'. | 17 |
| p1. ln 17 : Are we sure 'basic' is needed here (and in the following sentence). I suggest reducing the wording as much as possible to avoid confusion. | Changed to ,structures' following C1. | |

| | | |
|---|---|---|
| p1.Ln 22 : Suggestion : '..OC dynamics **along them ** is an essential and **relevant** focus on ocean research.' Instead of 'OC dynamics resulting from the multiplicity of these pathways and the human influence on them is an essential and very productive focus of ocean research'. As I am not sure the human influence is the main topic of this paper, and as I do not see how a focus can be productive. | We agree that the human influence is not our key point and shorten the sentence. However, we do not adapt the "along the pathways" as we argue that these dynamics are among others influenced by the interplay of pathways. This information would be lost. We change it to: Therefore, understanding marine OC pathways and the current and future marine OC dynamics resulting from the multiplicity of these pathways is an essential and very productive focus of ocean research. | 22-23 |
| p1. Ln 24-27 : I am not sure I understand the sentence. Is it the comprehensive observations and the sophisticated numerical models who improved the carbon budgets ? Maybe consider a rephrasing of the sentence. | We change it to: Comprehensive observations and sophisticated numerical models, e.g. by the Joint Global Ocean Flux Study..., improved carbon budgets... and quantitative estimates of the contribution of individual organisms ..., to name but a few, are continuously expanding our understanding of OC pathways and the marine OC cycle. | 23-25 |
| p2. Ln 29 : At this stage the definition of higher-level structures, core mechanisms is not intuitive. I suggest sticking with what will be used after ( Pathways and sequence of processes). | We do not use our terminology in this sentence, because we are not describing what we use or define here, but paraphrasing what other publications have used and done. However, we shortened the sentence to: ...generalise OC pathways as a sequence of processes or a core mechanism. | 29 |
| p2. Ln 29 - 31 : Suggestion : ' ..of the OC cycle, the studies focus only on the description of pathways related to the interest of the research''. Instead of '... OC cycle, these concepts have a relatively narrow focus and consider a selection of pathways.' | We changed it to: ...the OC cycle, these concepts only consider a selection of pathways related to the respective research focus. | 30-31 |
| p2 . Ln 31-34 : Similarly as comment for p1. Ln 24-27 , the utilization and referencing of the example make it hard to understand. Is it an enumeration, or one sentence only ? Maybe consider a rephrasing of the sentence. | We changed it to: For example, some studies conceptualise and generalise pathway structures for specific carbon pools e.g. dissolved OC in the microbial pump ..., for a selection of species such as bacteria in the microbial loop ... or for physical processes of different scales e.g. large-scale or eddy-subduction export ... | 31-34 |

| | | |
|---|---|---|
| p2. Ln 43 : Suggestion : ' useful' instead of 'plausible' ? | Changed it to: plausible and useful. | 43 |
| p2. Ln 48 : Why do we have 'graphics' twice in the sentence ? | Changed it to: within the respective graphics or compared to schemata in other publications | 48 |
| p2. Ln 50 - 53 : Suggestion : ' For example, Steinberg and Landry (2017), Cavan et al. (2019), Anderson and Ducklow (2001) and Boscolo-Galazzo et al. (2018), while aiming to represent the same pathways do not use the same visual representation leading to inconsistencies. As the aim of such studies is not to create congruent conceptual representations of the OC cycle, their visualizations are still useful tools to highlight their research focus in an overarching picture. ' | We changed it to:... visually detach processes from their products, such as DIC, or do not mention some products in the figures at all. As the aim of such studies is not to create congruent conceptual representations of the marine OC cycle, their visualizations are still useful tools to highlight their research focus in an overarching picture.

 Although we understand the referee's point, we cannot change the 2 sentences as suggested as the first suggested sentence would imply that there are only inconsistencies when comparing figures but our argument is that there are inconsistencies within single figures too. | 50-52 |
| p2. Ln 57 : Suggestion : ' Non-congruent graphics within the scientific literature to represent a same concept do not exploit the full potential' | We refrain from changing the sentence. As we argue that it is not only incongruence within different figures but also within one figure. The suggested sentence would be misleading. | |
| p2. Ln 58 : Do we want to stay large and talk to OC cycle, or do we want to specify 'marine OC cycle' in this explanation ? | Changed it to: marine. | 58 |
| p3. Ln 65 : Do we want to stay large and talk to OC cycle, or do we want to specify 'marine OC cycle' in this explanation ? | Changed it to: the marine OC cycle | 65 |
| p3. Ln 68 : I suggest removing core to avoid confusion on the definition associated with 'core similarity' that may not be clear at that stage of the manuscript. | We removed 'core'. | 66 |
| p3. Ln 73 : Do we want to stay large and talk to OC cycle, or do we want to specify 'marine OC cycle' in this explanation ? | We changed it to: marine. | 67 |
| p3. Ln 71-75 : I really appreciate this paragraph. It is well structured and gets straight to the objectives of this study. It is a nice addition to the first version of the manuscript. | We thank referee 2. | |

| | | |
|---|---|---|
| p3. Ln77and 78, 79 : I suggest removing core to avoid confusion on the definition associate with 'core similarity' that may not be clear at that stage of the manuscript. | We removed 'core'. | 77,78,79 |
| Space : To highlight that you are considering 'Atmosphère', 'Ocean' and 'Sediment' I suggest to list all your 5 spaces in the example column. Suggestion : Atmosphere Space (AS), Ocean spaces (e.g. Surface layer space (SLS) and Water column space (WCS)) ; Sediment spaces (e.g. Upper (USS) and Lower (LSS) sediment spaces). ' | While we see the referee's point, we have intentionally included only the spaces associated with the three example pathways at the top of the table. To emphasise that we are only providing examples connected to pathways 1-3, we change the heading to: 'Term' 'Definition' 'Examples based on pathways 1-3' | Table 1 |
| Initial position : Suggestion for the Example column, to use the same wording : .. in the ** Surface Layer Space** instead of surface space. | Agreed and changed. | Table 1 |
| For me the terms 'pool' , even intuitive, should be described as well as agent (not as intuitive) that you use several times in the abstract and Introduction. | We agree. Both are now included in Table 1. | Table 1 |
| My understanding is that 'pathway patterns' and 'sub-patterns' are the same thing. I suggest removing the 'sub-pattern' wording here and in the following text to avoid any confusion in the wording. | We agree and change the sentence following C1 to:
A structure that comprises all pathways returning to the initial position is named closed loops.
A structure that comprises all pathways not returning to the initial position is named 'open' loops. | Table 1 |
| I do not think the first line of the table with the Mapped example pathways in the base pathway concept is informative, it leads more to confusion in my point of view. | We have included these pathway examples intentionally to show how we have moved from single pathways with processes to structures with sequences of functional segments (see C1). We hope that by changing the heading to "Examples based on pathways 1-3" we have made the connection clearer. As we have noticed that the distinction between the terms still seems to be partly misleading, we change it according to C1. We will keep the pathways 1-3 on top of the table in any case, as they show what a pathway is and how a pathway merges into structure. | |
| p4. Ln 114-115 : This sentence is not useful or can be merged with the first one. | We changed it to: To this end, we generate a literature-based pathway concept (see Supplement A) by collecting and mapping the different pathways that an OC compound can | 112-114 |

| | "go" within the marine OC cycle based on a non-systematic literature review. | |
|---|---|---|
| p4. Ln 115-116 : Maybe you can try to have a logical order when listing the spaces : up to down (Atmosphere - surface - sediment) or down to up (sediment - surface - atmosphere) . | We restructured the sentence to highlight that the pathways either return or leave: The individual pathways in this concept are defined by sequences of processes (Table 1), such as sinking and remineralisation, and either return to the initial position in the surface water or leave the marine system to the sediment or the atmosphere. | 114-116 |
| p4. Ln 117 - 118 : This sentence is really hard to understand as a lot of things are mentioned with no clear definition or point of difference : 'base pathway', 'mapped pathways', 'core structures', 'core patterns of OC pathways'. It is really hard to get the nuance among the notions. Maybe the 'base' pathway concept can be named as 'litterature-based-pathway-concept', 'mapped pathways' which are the ones you are describing can be named simply 'OC pathways', and I do not get the sense and distinction of core structures/patterns. | We changed the sentence to: We compare the OC pathways in the literature-based pathway concept and condense their similarities into generally applicable structures.

In addition, we changed the name of the base pathway concept to literature-based pathway concept. | 116-117 |
| p6. Ln 123 : Following my previous comment, you can use the appropriate appellation ' To explain how to compare and condensed litterature-based-pathway-concept and define' | Changed it to: To explain how the pathways of the literature-based pathway concept can be compared and condensed to define structures of the marine OC cycle, we… | 120-121 |
| p6. Ln 123 : Once more, what is the 'core patterns' meaning ? | We deleted it here. | 123 |
| p6. Ln 140 : I do not get the meaning of the 'entire-city-beach route'. Following your explanation it should be named ' harbor front beach route' otherwise I do not get why the harbor front beach route is a subordinate of the entire-city-beach route' as it is the same thing.. ? | We thank referee 2 for this question. We adapted the description and explanation as it was indeed partly misleading. | 120 and following paragraphs |
| p6 Ln 144 : Please be consistent in the wording. What route is referring to here ? Path segments or Pathway patterns ? | We changed it to: One could for example also distinguish other structures based on the method of crossing the lagoon or find further differences and commonalities between the pathways in the rest of the city and define additional structures. | 146-148 |
| p6. Ln123-147 : From the explanation you provide, I drew a schematic (Schematic 1). But it seems that the term 'pathways' is not properly placed in my schematic. | We thank referee 2 for the schematic and the time invested into the review. We adapted our terminology (C1) and | 120 and following paragraphs |

| | | |
|---|---|---|
| Maybe it is my understanding wrong, or maybe something is misleading in the explanation. I'll let you have a second look on the text to be sure. | streamlined the explanations throughout the text. | |
| p8. Ln188 : For consistency with my comment about Table 1. 'sub-patterns' should be replaced by 'pathways'. | Deleted sub-pattern- see C1. A clarification: A pathway is always an individual sequence of processes. Example: Pathway 1: travel to the port via road A and take the public ferry. It can also be described as a sequence of functional segments if we transfer the processes to their general function. For example: Pathway 1: get to the harbour and cross the lagoon. A structure is always a condensation of several pathways. Different structures (of different hierarchical order) can be defined depending on the resolution of details.  For example, the rDOC loops belong to the structure closed loops. Or in the analogy, the " behind the harbour front beach" structure belongs to the "the entire city beach" structure. | |
| p8. Ln 221 : At the end of the sentence, Are we sure the wording is sub-pathway patterns and not 'pathway patterns' ? | We changed it to: However, users of the concept can identify and combine other functional segments to define different higher-resolution structures. | 223-224 |
| p9. Ln 229 : for consistency it should be 'pathways' and not 'sub-patterns' (See comment on Table 1). | Changed to: We define four structures of 'open' loops. | 226 |
| p7. Ln 160 : As you already said in Sect 2 this, I recommend using wording such as ' As previously mentioned, the path segments…' . | We slimmed down the paragraph. As a result, the relevant passage has been omitted. | 163 ff. |
| p10 Ln235 to p16 Ln 330 : For consistency with the italic used to characterize pathway and path segments, can we place the processes in italic in the text too ? | We do not put the processes in italics because we do not define them, but only compile them. | |
| p12 Ln. 253 : 'A of (r) DOC in 2)' . Does the '2' refer to Fig.2 ? | Indeed. We changed it to: A of (r)DOC in Figure 2 | 250 |
| p15 Ln. 287 : I suggest the reading of this paper to add a reference here :Goldthwait, S., Yen, J., Brown, J., and Alldredge, A.: Quantification of marine snow fragmentation by swimming euphausiids, Limnol. Oceanogr., 49, 940–952, https://doi.org/10.4319/lo.2004.49.4.0940 , 2004 | We did as recommended. | 281 |

| | | |
|---|---|---|
| p15 Ln. 294 : The wording here may be misleading. 'sub-patterns' should be pathways, and POC-DOC remineralisation 'sub-loops' ? | A higher-level structure, comprises several levels of lower-level structures. For example, closed loops are the most superordinate structure in the marine OC cycle. rDOC and remineralisation loops belong to these closed loops. The POC-DOC remineralisation loop belongs to the remineralisation loops and closed loops. The more details are included and the higher the resolution, the more the structure resembles the individual pathways up to the point where a pathway is described rather than a structure. We adapted the description following C1. | |
| p16 Ln. 325 : Instead of sub-pattern, shouldn't it be 'sub-loop' ? (See my schematic 2). | Changes were made following C1. | |
| p16 Ln.327 and 329 : Does the term 'sub-patterns' refer here to the pathways or to the sub-loops previously mentioned and called-sub-patterns before ? | Changes were made following C1. | |
| p16 Ln. 330 : Does the term 'patterns' refer here to pathway patterns, or the previous term sub-patterns that are confusing (see previous comment) ? | Changes were made following C1. | |
| For consistency it should be 'pathways' and not 'sub-patterns' (See comment on Table 1) in the | Changes were made following C1. | |
| Table 2 : Uniformize the term 'sub-pattern/Pathways' (See previous comments on that point). | Changes were made following C1. | |
| Table 3 : It is a great improvement of the first table 3 proposed in the first version, congratulations ! | We thank referee 2 for this comment. | |
| May I suggest to remove the repetition of the column names (Process, loop syntax, etc.) for each Path segment. The reader may refer to the first one if s/he needs a reminder. Therefore I suggest to place the column names above the first path segments to be clear these names apply for the entire long-table. | While we understand the referee's point of view, we tested this version in the first review round and it did not improve readability. Certainly, the table becomes shorter this way, but from our point of view at the expense of clarity. | |
| I am wondering if we do not have the same information twice with Fig 2 and Table 3 ? Do we want to keep both, or do we want to choose one of them ? Just a thought | We think that the table adds additional and relevant information and thus keep both. | |

| | | |
|---|---|---|
| Entire discussion : To avoid the repetition of the main reference Giering and Humphrey 2020, maybe in the second paragraph of the discussion you can make a statement that mentions that in the following analysis the description of BCP used as reference is based on Giering and Humphrey 2020 ? With this statement the reader will know that further assumption will refer to their work and you would not have to mention it in every paragraph ? | We thank the referee for this very helpful comment. We decreased the repetition of the reference and added a footnote saying: If not mentioned differently, we always refer to the BCP definition by ... in the following discussion. | 336 |
| p17 Ln 342 : This sentence is a repetition of the previous paragraph. It should be reswamp if you want to keep the information that relates with your example (F [SLS]). | We changed it to: Using the syntax of our concept, the defined BCP involves the uptake of inorganic carbon into biomass in the surface waters (F [SLS]) and the OC position change to the interior of the ocean (A [Ocean Interior]), where it is remineralised to DIC (D [Ocean interior]) (Figure 3 panel (a)). | 340-342 |
| p17 Ln 351 and 352, 353, 357, 358 , 360: As mentioned before, sub-patterns should be removed and sub-loop should be used (See my schematic 2). | Changes were made following C1. | |
| p17 Ln 354 and 358 : Does 'loop' refer to the sub-loop ? If yes please use consistent wording. | Changes were made following C1. | |
| p17 Ln346 to 363 : It is not really clear when reading the text and having the Figure 3 under the eyes how the number of loops is determined. In p17 Ln. 348 It is mentioned 'to close the loop' inducing that there is one loop in the panel (a) is misleading of what it is stated at p17 Ln 354 when ' only two loops of the superodinate loops of panels (a)'. Is it possible to have the number of loops associated with the pathway patterns mentioned in the Figure 3 legend ? Similarly, the loops are not easy to see on the Figure pannels, and when the author refers in the text to seven loops or six loops form panels (c) and (d) it is misleading with the numbering of pathway patterns mentioned in the Figure. I suggest either talking only of pathway patterns numbers in the text to fit with the legend of the Figure, or to switch the legend of the figure with the numbering of loops to fit with the text. | We agree that it was partly hard to follow the comparison of the numbers and thus now directly address the numbers of the pathways in figure 3. In addition, we shortened the paragraph following also a comment by referee 1. | 346-358 |

| | | |
|---|---|---|
| Figure 3:The space (SLS) and (WCS) can be placed once on the left side of the figure. | We changed it accordingly. | Figure 3 |
| Figure 3: The term 'Processes' can be placed in bold above biota-induced and physical | Following comments of referee 1, we added: a) BCP as defined, b) Resolved pools c) Direct biota-induced transport and c) Physical processes** as titles for the panels. | Figure 3 |
| Figure 3: The term 'loops' can be placed in bold above the various loops specified | As loop is only a name of a structure and we want to have it more inclusive (plus the original definition of the BCP misses the path segment E to form a loop), we use structure instead of loops. | |
| Supplement B, In the box 6, as Pathway pattern abbreviation has already been described in box 4 you can either use the full wording or the abbreviation only but not both, it is confusing. | We changed it accordingly and also adapted the description a bit to account for C1. | |
| Editorial/Typo comments : | | |
| p8 Ln 189 : Do not use italic for and between the two sub-patterns/pathways. | Changed accordingly. | 186 |
| p10. Ln 232 : Do not use italic for and between the two sub-patterns/pathways. | Changed accordingly. | 226 ff. |
| Legend Figure 1 : "loop" when talking about srDOCL and LrDOCL shouldn't be plural ? | We checked the legend but couldn't find a mismatch. | |
| Legend Table 2 : "loop" when talking about srDOCL and LrDOCL shouldn't be plural ? | We checked the legend but couldn't find a mismatch. | |
| p12 Ln. 249 : Even if it is the beginning of the sentence, I suggest to force the r of (R)DOC to be in lowercase. | We changed it as recommended. | 246 |
| p12 Ln 253 : The path segment A should be placed in parenthesis. | We put the letters in parenthesis only after writing out the functional segment. E.g. 'remineralisation of OC (D) is involved in' versus 'functional segment D is involved in…' | |
| p12 Ln 258 : The path segments A and E should be placed in parenthesis. | same as above | |
| p12 Ln259 + all the manuscript+Figures/Tables : Shouldn't be '(r) DOC' instead of rDOC ? Maybe I am confusing the meaning, but please review all the manuscripts and supplementary material if the wording with and without parenthesis means the same thing. If not please mention somewhere the difference between the two ways of writing it. | We checked the manuscript and added that (r)DOC means a process is valid for (DOC and rDOC). | 246 |
| p12 Ln 267 : The path segment A should be placed in parenthesis. | same as before | |
| p12 Ln 272-273 : The path segments A and E should be placed in parenthesis. | same as before | |

| | | |
|---|---|---|
| p12 Ln 277 : The path segment D should be placed in parenthesis. | same as before | |
| p15 Ln 281 : The path segment D should be placed in parenthesis. | same as before | |
| p16 Ln 303 : The path segment D should be placed in parenthesis. | same as before | |
| p17 Ln 348 : The path segment E should be placed in parenthesis. | same as before | |
| p17 Ln 351 and 354 : The path segment E should be placed in parenthesis. | same as before | |
| Figure 3 : | | |
| - I wonder if the figure 3 would be better if seen as landscape instead of portrait within the page ? | Since the figure placing is usually decided by the technical editor, we have not changed the orientation at this point in time. | |
| p19 Ln385 : The path segment E should be placed in parenthesis. | same as before | |
| **Supplement:** | | |
| First Review the arrow legends, as some are placed below the arrows and are sometime difficult to read (e.g. Coastal in the sediment part 'Consumed macrophytes' below the black arrow) ; | First of all, we would like to thank referee 2 for taking so much time and reviewing the supplement. We have tried to take some of the feedback into account, but as it is a supplement we have not been able to take all the referees' comments into account. We have corrected spelling errors where we found them and changed some of the label placements. | |
| Some of the text are missing space between words (e.g. Coastal 'Carnivoresand detritivores') | Regarding the questions about why we do not use boxes for bacteria, viruses and faecal pellets: We use boxes for bacteria and viruses. Faecal pellets go into the POC pool. The arrows with bacteria, viruses and faecal pellets indicate that there can be a change in position between pools of different water layers. We distinguish between benthic carnivores and benthos that are not carnivores. | |
| Some of the arrow descriptions are similar, maybe you can manage to have the same infos placed where the arrows merge ? (e.g. Coastal, Sinking of resting stages). | | |
| Why don't' you use boxes for Fecal pellets,bacteria and Virus ? | | |
| As you refer to 'benthic carnivores', what imply 'Benthos' ? | | |
| Why only referring to mammals ? You may refer to the upper trophic levels to be as general as possible ? | Since we include carnivorous fish, mammals are the only group of higher trophic levels left from our point of view. | |
| - Shouldn't 'Pahotrophy' be Phagotrophy ? | | |
| Physical Transport is written numerous times (with some typo (Phyisical)), but it is already linked to blue arrows that are mentioned in the legend as physical-induced, so maybe there is no need to write it down ? | We retain physical transport, although we agree that it is very often included. However, we feel that the size of the concept does not really allow us to delete information, as it would take time to find the information elsewhere. | |
| - Does 'Autigenic' shouldn't be Authigenic ? | | |

| | | |
|---|---|---|
| Is it possible to have this huge diagramm 'interactive' ? Is it possible to have in the legend the SLRL loop display for example, and when someone click on it is only the SLRL arrows and boxes and infos that appear for a better visualization? | The concept is partly interactive in the sense that it shows the references and short descriptions when you scroll over the arrows. However, we think that implementing an interactive level as suggested by reviewer 2 would be great. However, this was not within the scope and time frame of our project. | |